# PRO-TRANS: PROGRESSIVE TENSOR RING WITH ATTENTION GUIDED LOCAL SMOOTHING REGULARIZATION

## ABSTRACT

The generalization of adversarial defense methods remains a critical open challenge, and optimization-based adversarial purification methods employing tensor network representations have recently shown strong potential. However, such tensor-based defense methods operate solely on the given input without relying on prior knowledge, which inevitably leads to overfitting to adversarial perturbations. Moreover, their iterative optimization procedures incur substantial computational overhead during inference. In this paper, we propose Pro-Trans, a novel tensor-based adversarial purification method that integrates progressive tensor ring with attention guided local smoothing regularization. Specifically, our progressive tensor ring avoids redundant upsampling operations, thereby reducing computational overhead and accelerating convergence. In addition, the proposed regularizer adaptively applies varying degrees of local smoothing regularization across different regions, thereby suppressing perturbations while mitigating semantic loss. Experimental results show that Pro-Trans consistently outperforms existing methods across diverse adversarial settings on CIFAR-10, CIFAR-100, and ImageNet, achieving state-of-the-art performance while maintaining low computational cost. The code will be available upon acceptance.

## 1 INTRODUCTION

Deep learning models have achieved remarkable success in diverse applications, yet their performance degrades sharply under adversarial attacks (Szegedy et al., 2013; Goodfellow et al., 2015). To counter such risks, numerous adversarial defense methods have been proposed, and they can be divided into two main categories: adversarial training (AT) and adversarial purification (AP). AT enhances robustness by retraining models against known attacks but often struggles to unseen ones (Laidlaw et al., 2020). AP introduces a purifier before the classifier to remove perturbations, generally achieving better transferability than AT, though its reliance on pretrained generators limits adaptability to new datasets and increases computational cost (Nie et al., 2022; Lin et al., 2023).

Recent tensor-based AP methods have demonstrated stronger defense generalization compared to prior approaches (Yang et al., 2019; Entezari & Papalexakis, 2022; Bhattarai et al., 2023; Lin et al., 2025). This advantage stems from their optimization-based nature, which avoids reliance on pretrained generators, specific datasets, or fixed model parameters, thereby reducing vulnerability to both white-box and black-box attacks. Consequently, tensor-based AP holds great promise for defense generalization performance under diverse adversarial settings. Despite the above advantages, certain intrinsic properties of tensor networks (TNs) hinder the further improvement of TN-based AP methods. In particular, TNs tend to reconstruct both semantic image details and adversarial perturbations, increasing the risk of overfitting to adversarial examples. To mitigate this, Lin et al. (2025) propose a novel TN-based AP method (TNP), which integrates upsampling, downsampling, and adversarial optimization process, thus better removing perturbation and demonstrating promising defense generalization performance across diverse adversarial settings. However, it still incurs high computational overhead, as noted by Lin et al. (2025). In addition, the adversarial optimization process makes the optimization unstable, which may result in ineffective purification or even amplify perturbations (Goodfellow et al., 2014; Salimans et al., 2016). Collectively, achieving efficient and stable adversarial purification with TNs still remains a challenging problem.

To address the aforementioned challenges, we propose Pro-Trans, a novel tensor network-based adversarial purification method that integrates Progressive Tensor Ring (PTR) with Attention-Guided Local Smoothing Regularization (AGLSR). Unlike traditional coarse-to-fine optimization strategies that progressively modify the structure of tensor network, our proposed PTR progressively modifies optimization objectives within a fixed, pre-defined Tensor Ring structure. This design eliminates the need for interpolation-based upsampling and avoids the computational overhead of dynamic structural changes, which is commonly used in conventional coarse-to-fine TNs (Loeschcke et al., 2024; Lin et al., 2025), resulting in a more efficient optimization. Moreover, by removing the instability introduced by interpolation-based upsampling, PTR achieves more stable convergence. Finally, the coarse-stage optimization naturally provides better parameter initialization for subsequent finer stages, further enhancing both efficiency and stability of the purification process.

Additionally, recent studies have shown that the damage of adversarial perturbations increases monotonically with frequency, whereas low-frequency structures and image contents remain relatively unaffected (Pei et al., 2025). Besides, the natural images not only exhibit low-rank structures but also adhere to the local smoothness prior (Lan et al., 2023). Naively applying smoothing techniques may lead to over-smoothing, thereby degrading essential semantic information. Motivated by these observations, we introduce AGLSR, specifically designed to further improve purification performance in PTR. AGLSR adaptively applies varying degrees of local smoothing regularization across different regions, guided by the model's attention. This mechanism effectively suppresses perturbations while minimizing semantic distortion, thereby enhancing the purification quality.

To demonstrate the effectiveness of our proposed method, we conduct extensive empirical experiments on three benchmark datasets, comparing its performance against state-of-the-art defense methods under diverse attack settings, including cross-dataset, cross-threat, and cross-attack scenarios. The results show that our approach achieves competitive robustness with mainstream methods while exhibiting superior defense generalization performance. Furthermore, convergence analysis of the PTR highlights its faster and more stable optimization compared with existing approaches. Finally, ablation studies on Pro-Trans confirm that PTR substantially reduces computational overhead, while AGLSR effectively balances perturbation suppression and semantic detail preservation. In general, our contributions can be summarized as follows:

- We propose the first coarse-to-fine Progressive Tensor Ring (PTR) for AP. By avoiding interpolation-based upsampling and progressively freezing/unfreezing core tensors, PTR achieves significantly faster and more stable convergence than existing TNs.

- We design an Attention-Guided Local Smoothing Regularization (AGLSR) that integrates total variation with attention masks to adaptively balance semantic preservation and perturbation suppression, thereby effectively reducing over-smoothing.

- By combining PTR and AGLSR, we establish the first TN-based AP framework that simultaneously improves efficiency, stability, and defense generalization, directly addressing the core limitations of prior TN-based AP methods.

- Extensive experiments on various datasets demonstrate that Pro-Trans achieves state-of-the-art robustness while significantly reducing computational overhead. Ablation studies and visualizations further highlight the individual contributions of PTR and AGLSR.

## 2 RELATED WORKS

### 2.1 EXISTING PARADIGMS OF ADVERSARIAL DEFENSE

To counter the impact of adversarial attacks and improve the robustness of deep learning models, research has mainly focused on AT and AP. AT enhances robustness against known attacks by retraining the model with adversarial examples incorporated into the training set (Goodfellow et al., 2015). TRADES (Zhang et al., 2019) introduce explicit loss functions that balance standard accuracy (SA) and robust accuracy (RA). Wong et al. (2020) propose approaches to reduce computational overhead. However, AT still suffers from poor generalization when facing unseen attacks or new datasets. In contrast, AP methods insert a purification module before the classifier, leveraging pre-trained generative models to project adversarial examples back onto the benign data manifold (Samangouei et al., 2018; Yoon et al., 2021; Nie et al., 2022). Compared to AT, AP tends to perform better against

unseen attacks, but this capability is usually confined to the distribution of the training data used for the generator, limiting its ability to generalize to new distributions. In general, AT and AP together constitute the main paradigms of adversarial defense, but both exhibit limited generalization across diverse attack scenarios and introduce additional computational costs.

## 2.2 TENSOR-BASED DEFENSE METHOD

To address the limitations of existing defense methods, tensor-based adversarial purification methods have recently emerged as a promising research direction. As a long established tool in signal processing, TNs include Tensor Train (TT, Oseledets, 2011), Quantized Tensor Train (QTT, Oseledets, 2009), Tensor Ring (TR, Zhao et al., 2016), Quantized Tensor Ring (QTR, Zhao et al., 2016), and PuTT (Loeschcke et al., 2024). By leveraging low-rank property and multi-dimensional structure representations, TNs enhance robustness from the perspective of data purification. ME-Net employs matrix estimation to reconstruct sampled images and disrupt adversarial perturbation structures (Yang et al., 2019). TensorShield utilizes tensor decomposition to approximate inputs with low-rank representations, filtering out high-frequency noise (Entezari & Papalexakis, 2022). TNP adopts downsampling, upsampling, and an adversarial optimization process, exploiting the TN's ability to remove Gaussian noise, thereby achieving stronger generalization ability when facing diverse attack scenarios (Lin et al., 2025). Nonetheless, these approaches still face challenges in simultaneously reconstructing image details and removing perturbations, as well as inefficiencies in TN optimization. In contrast, we aim to design a tensor-based AP method that exploits the intrinsic low-rank and local smooth properties of natural images through efficient TN, thereby enhancing both generalization and efficiency while significantly reducing computational overhead.

## 3 METHOD

This section presents Pro-Trans, which combines PTR and AGLSR. PTR is a novel TN that performs coarse-to-fine optimization without redundant upsampling by progressively adjusting optimization objectives and participating core tensors. AGLSR is a regularizer that adaptively suppresses perturbations while preserving semantics. We next detail each component before introducing Pro-Trans.

## 3.1 PROGRESSIVE TENSOR RING

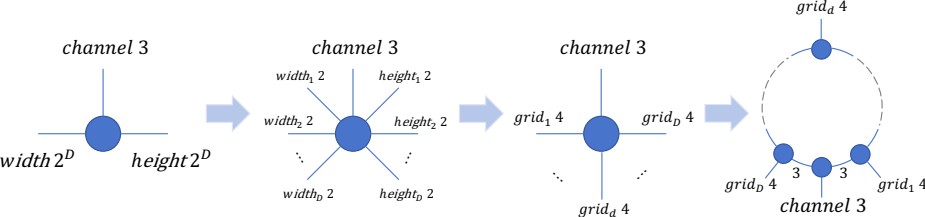

Figure 1: Tensorization and decomposition of an RGB image: height and width are decomposed into size-2 height and width modes and grouped into size-4 grid modes from coarse to fine, and finally decomposed by TR.

Although existing coarse-to-fine TNs are effective at purifying Gaussian-like noise at coarse stages (Lin et al., 2025), their design typically relies on interpolation-based matrix product operators (MPOs) for upsampling and TT-SVD (Oseledets, 2011) to prevent exponential rank growth, both of which introduce considerable computational overhead and negatively impact the stability and convergence of the TN-based purification process, as previously discussed.

To address these challenges, we propose Progressive Tensor Ring (PTR) to achieve coarse-to-fine TN optimization without the extra upsampling steps, thereby enhancing the stability and efficiency of the optimization process. As shown in Figure 1, an RGB image can be represented as a third-order tensor $\mathbf{X} \in \mathbb{R}^{H \times W \times C}$, where $H$, $W$, and $C$ denote the height, width, and channel dimensions, respectively. For simplicity, we assume that $H = W = 2^D$, where $D$ represents the maximum decomposition depth. To enable coarse-to-fine optimization, we perform a quantization on the spatial

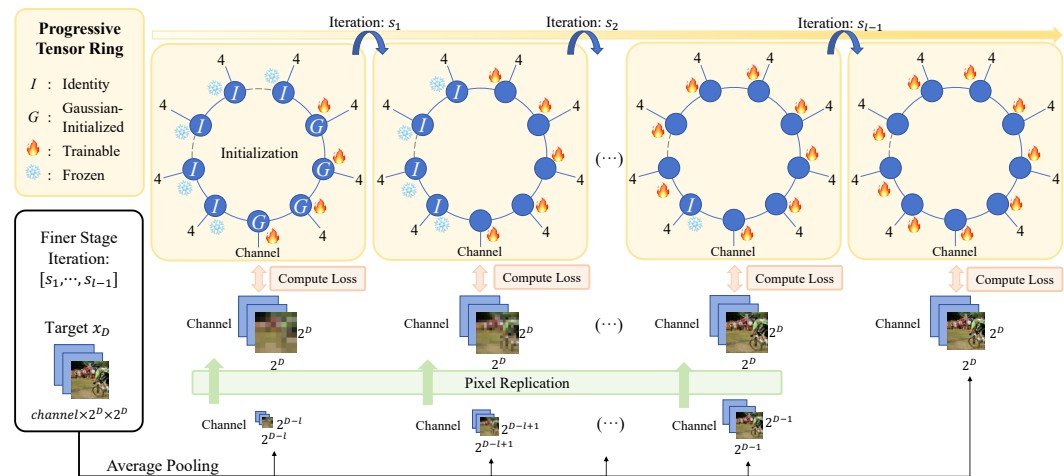

Figure 2: Illustration of PTR. Optimization targets are obtained by average pooling and pixel replication. PTR is initialized with Gaussian-distributed parameters for the first $D - l$ cores and the channel core, while the remaining cores are set as identity tensors. At each stage $d$, PTR optimizes the first $d$ cores and the channel core with target $x_d$, then progressively shifts to higher resolutions by unfreezing additional cores in a coarse-to-fine manner.

dimensions (height and width) as:

$$H = \prod_{k=1}^{D} h_k, \quad W = \prod_{k=1}^{D} w_k, \quad h_k = w_k = 2. \tag{1}$$

At each scale $k$, the pair $(h_k, w_k)$ forms a grid mode, yielding a hierarchical four-dimensional representation from coarse to fine granularity (Oseledets, 2009): $\mathbf{X}^{(q)} \in \mathbb{R}^{\text{grid}_1 \times \text{grid}_2 \times \cdots \times \text{grid}_D \times C}$. We then adopt the TR to factorize the input into a channel factor and multiple grid factors:

$$\mathbf{X}^{(q)}_{g_1, g_2, \ldots, g_D, c} \approx \text{tr}\Big(\mathbf{G}^{(1)}[g_1]\mathbf{G}^{(2)}[g_2] \cdots \mathbf{G}^{(D)}[g_D]\mathbf{G}^{(C)}[c]\Big),$$
$$\mathbf{G}^{(d)} \in \mathbb{R}^{r_{d-1} \times n_k \times r_d}, \quad n_k = \dim(\text{grid}_k) = 4, \quad \mathbf{G}^{(C)} \in \mathbb{R}^{r_D \times C \times r_0}, \tag{2}$$

where $r_d$ denote the adjacent rank that connect the $d$-th and $(d + 1)$-th cores in the TR, and the circular constraint $r_0 = r_{D+1}$ closes the ring structure. Unlike other TNs, the ring structure enables TR to achieve stronger expressive power and higher parameter efficiency, providing a symmetric representation that avoids the boundary constraints inherent in TT (Zhao et al., 2016). With the PTR topology formally defined, we next describe its optimization process, which follows a coarse-to-fine strategy to progressively refine reconstruction while ensuring stability and efficiency.

As shown in Figure 2, we first apply average pooling to downsample the target image, and then upsample it back to the original resolution by pixel replication, therefore constructing optimization targets at different coarse-to-fine stages. Before optimization starts, the core tensors in PTR are initialized by combining Gaussian random initialization with identity tensors $\mathbf{I}$, as follows:

$$\mathbf{G}^{(k)} \sim \mathcal{N}(0, \sigma^2), \ k = 1, \ldots, d; \quad \mathbf{G}^{(C)} \sim \mathcal{N}(0, \sigma^2); \quad \mathbf{G}^{(k)} = \mathbf{I}, \ k = d+1, \ldots, D. \tag{3}$$

More details about the identity tensor can be found in the Appendix A.2. In this way, PTR reconstructs images at resolution $2^d$ using the first $d$ cores, while the remaining cores $\{\mathbf{G}^{(i)} \mid i = d+1, d+2, \ldots, D\}$ are set as identity tensors to replicate pixels. This design naturally aligns with the previously described pixel-replication upsampling used to match low-resolution optimization targets to the original resolution. Thus, at $d$-th optimization stage, the optimization target is the image $x_d$ at resolution $2^d$, and the optimization variables are the first $d$ core tensors $\{\mathbf{G}^{(i)} \mid i = 0, 1, \ldots, d\}$ and the channel core tensor $\mathbf{G}^{(C)}$, while the other core tensors remain frozen. Once the optimization at stage $d$ is completed, the resolution is increased to $2^{d+1}$. The optimization target then shifts to $x_{d+1}$ at resolution $2^{d+1}$, with the $(d+1)$-th core tensor unfrozen and incorporated into the set of

Figure 3: Overview of the proposed Pro-Trans. Multi-resolution targets are first generated through average pooling and pixel replication, and the attention mask is derived from the downstream classifier. PTR then performs coarse-to-fine optimization: in the coarse stage it captures long-range structures and suppresses Gaussian-like perturbations, while the effect of AGLSR becomes increasingly dominant in the fine stage, suppressing perturbations and reducing the loss of semantic details.

optimization variables. Formally, the optimization at stage $d$ can defined as:

$$\min_{\{\mathbf{G}^{(i)}\}_{i=1}^{d}, \mathbf{G}^{(C)}} L_d = \|X_d - Y_d\|_2, \quad \text{s.t. } Y_d = \text{TR}\big(\mathbf{G}^{(1)}, \ldots, \mathbf{G}^{(d)}, \underbrace{\mathbf{I}, \ldots, \mathbf{I}}_{D-d}, \mathbf{G}^{(C)}\big). \quad (4)$$

where $X_d$ is the downsampled target and $Y_d$ is given by the contraction of PTR. This progressive procedure is repeated until the final stage, where the optimization target is the input image and all core tensors are jointly optimized. Overall, PTR provides a novel coarse-to-fine optimization scheme that avoids complex upsampling and instead achieves flexibility by progressively adjusting the optimization targets and the set of trainable core tensors.

## 3.2 ATTENTION-GUIDED LOCAL SMOOTHING REGULARIZER

Adversarial purification aims to remove perturbations while preserving semantic content. Coarse-to-fine TNs first reconstruct low-frequency structures, where downsampled perturbations resemble Gaussian noise and can be effectively removed (Lin et al., 2025). As optimization proceeds to high-frequency details, perturbations with stronger destructive effects are inevitably reconstructed, leading to unsatisfying purification effect. Natural images exhibit the local smoothness prior: pixels change smoothly in flat regions but sharply at edges and textures (Lan et al., 2023). Leveraging this property, we introduce a local smoothing regularizer to encourage purified results to align with the natural image manifold and suppress high-frequency perturbations.

Among various smoothing technique such as Gaussian blur, Laplacian smoothing, or bilateral filtering, most either overblur semantic content or introduce optimization challenges. In contrast, Total Variation (TV) regularization penalizes the $l_1$ norm of image gradients, offering a mathematically simple and optimization-friendly formulation (Chambolle, 2004) that has been widely applied in image denoising and restoration (Wang et al., 2017). Unlike quadratic penalties such as Laplacian smoothing that overly suppress edges, TV effectively reduces local noise while preserving sharp edges and structural details. Thus, we adopt TV as our regularizer, yielding the following objective function: $L_d = \|X_d - Y_d\|_2 + \alpha \cdot TV(Y_d)$, where $\alpha$ controls the strength of the regularization.

To avoid semantic degradation caused by smoothing regularization, we further introduce an attention-guided mechanism that adaptively adjusts the regularization strength across regions. Specifically, we extract activation maps from different blocks of the downstream classifier, aggregate each across channels, upsample each to the input resolution and normalize each. The final attention

mask $M$ is then obtained by taking the maximum value across all blocks:

$$M_{i,j} = \max_{m \in [M]} \varnothing \big( \text{Bi}(\sum_{c=1}^{C_m} |f_m(Y)_c|^2) \big)_{i,j}, \quad \forall i \in [H], \forall j \in [W], \tag{5}$$

where $f_m(Y)$ denotes the activation maps of the $m$-th block, $C_m$ denotes the number of channel, $\text{Bi}(\cdot)$ denotes bilinear upsampling, and $\varnothing(\cdot)$ denotes normalization. The final objective function is:

$$L_d = \|X_d - Y_d\|_2 + \alpha \cdot TV(M \odot Y_d). \tag{6}$$

### 3.3 PRO-TRANS FRAMEWORK

Based on PTR and AGLSR, we propose a novel TN-based AP method, termed Pro-Trans. Leveraging the efficient and stable convergence of PTR, we employ it as the backbone for adversarial purification. To mitigate the tendency of tensor-based AP methods to restore perturbations, AGLSR introduces attention-guided, region-dependent local smoothing. As shown in Figure 3, the pipeline first constructs multi-resolution targets via average pooling and pixel replication, then derives an attention mask from the downstream classifier. PTR is initialized to perform coarse-to-fine optimization, with the loss defined in Equation 6. In the coarse stage, PTR primarily models long-range structures of the image, while the perturbations, transformed into Gaussian-like noise by average pooling, can be easily removed by TNs (Lin et al., 2025). In the fine stage, the role of AGLSR becomes increasingly prominent, imposing varying degrees of local smoothing regularization across regions to suppress perturbations while reducing semantic loss.

## 4 EXPERIMENTS

### 4.1 EXPERIMENTAL SETUP

**Dataset and Model Architecture.** We conduct experiments on three benchmark datasets: CIFAR-10, CIFAR-100 (Krizhevsky & Hinton, 2009), and ImageNet (Deng et al., 2009). For classification tasks, we employ ResNet (He et al., 2016) and WideResNet (Zagoruyko & Komodakis, 2017) architectures, using pretrained model weights provided by RobustBench (Croce et al., 2021).

**Adversarial Attacks.** We evaluate the defense performance of Pro-Trans against mainstream adversarial attack methods. AutoAttack was chosen as a well-established benchmark (Croce & Hein, 2020). In addition, we also test Pro-Trans under PGD (Madry et al., 2018) with EOT (Athalye et al., 2018) attacks, following Lee & Kim (2023).

**Implementation Details.** Due to the high computational cost of the experiments, we randomly selected 512 images from the test set for robustness evaluation, following Nie et al. (2022). All experiments are conducted on an NVIDIA RTX 4070 Ti Super GPU with 16 GB of memory, using CUDA version 12.6 and PyTorch (Paszke et al., 2019) version 2.8. For more implementation details, please refer to the Appendix A.3.

### 4.2 COMPARISON OF DEFENSE PERFORMANCE

Following the RobustBench protocol, we evaluated Pro-Trans using AutoAttack with $l_2$ and $l_\infty$ threats on CIFAR-10, CIFAR-100 and ImageNet, and compared its performance with other reported methods. As shown in Table 1 to 4, the results indicate that our method performs on the same level as the mainstream approaches, while achieving improvements of 1.56% in RA on CIFAR-10, 0.77% on CIFAR-100, and 8.40% on ImageNet compared to the TNP method (Lin et al., 2025). Following the experimental setting of TNP, we also observed that the standard WideResNet-28-10 suffers from overfitting to the limited dataset, which prevents Pro-Trans from achieving satisfactory performance. To address this, we also performed experiments with a robust classifier (Cui et al., 2024). Compared to using a robust classifier alone, our method still achieves an additional improvement of 5.47% in RA against AutoAttack $l_\infty$ threat ($\epsilon = 8/255$) on CIFAR-10. Overall, these results demonstrate that our approach holds considerable promise in enhancing robustness and further reveal the potential of tensor-based AP methods. In the tables, $^\dagger$ indicates the usage of additional synthetic images and $^*$ indicates the usage of the robust classifier.

Table 1: Standard and robust accuracy (%) against AutoAttack $l_\infty$ threat ($\epsilon = 8/255$) on CIFAR-10 with WideResNet-28-10 classifier.

| Defense | Extra data | SA | RA |
|---|---|---|---|
| Gowal et al. (2020) | ✓ | 90.82 | 60.55 |
| Pang et al. (2022) | ×[†] | 88.87 | 60.94 |
| Wang et al. (2023) | ×[†] | 93.16 | 68.36 |
| Cui et al. (2024) | ×[†] | 93.16 | 68.55 |
| Nie et al. (2022) | × | 89.02 | 70.64 |
| Lin et al. (2025)* | × | 91.99 | 72.85 |
| Ours | × | 82.42 | 59.76 |
| Ours* | × | 87.69 | 74.02 |

Table 2: Standard and robust accuracy (%) against AutoAttack $l_2$ threat ($\epsilon = 0.5$) on CIFAR-10 with WideResNet-28-10 classifier.

| Defense | Extra data | SA | RA |
|---|---|---|---|
| Rebuffi et al. (2021) | ×[†] | 92.77 | 79.69 |
| Rony et al. (2019) | × | 88.45 | 68.75 |
| Ding et al. (2019) | × | 88.87 | 65.43 |
| Nie et al. (2022) | × | 91.03 | 78.58 |
| Lin et al. (2025)* | × | 91.99 | 79.49 |
| Ours | × | 82.42 | 69.92 |
| Ours* | × | 87.69 | 81.05 |

Table 3: Standard and robust accuracy (%) against AutoAttack $l_\infty$ threat ($\epsilon = 4/255$) on ImageNet with ResNet-50 classifier.

| Defense | Extra data | SA | RA |
|---|---|---|---|
| Wong et al. (2020) | ×[†] | 54.49 | 27.15 |
| Engstrom et al. (2019) | × | 64.45 | 32.81 |
| Salman et al. (2020) | × | 66.99 | 38.28 |
| Nie et al. (2022) | × | 67.79 | 40.93 |
| Chen & Lee (2024) | × | 70.90 | 44.92 |
| Lin et al. (2025) | × | 65.43 | 42.77 |
| Ours | × | 64.84 | 51.17 |

Table 4: Standard and robust accuracy (%) against AutoAttack $l_\infty$ threat ($\epsilon = 8/255$) on CIFAR-100 with WideResNet-28-10 classifier.

| Defense | Extra data | SA | RA |
|---|---|---|---|
| Hendrycks et al. (2019) | ✓ | 59.23 | 28.42 |
| Rebuffi et al. (2021) | ×[†] | 59.77 | 33.01 |
| Pang et al. (2022) | ×[†] | 61.52 | 32.03 |
| Wang et al. (2023) | ×[†] | 71.29 | 38.28 |
| Cui et al. (2024) | ×[†] | 72.85 | 39.45 |
| Lin et al. (2025)* | × | 62.30 | 44.34 |
| Ours* | × | 65.62 | 45.11 |

## 4.3 Comparison of Defense Generalization under Diverse Attack Scenarios

To evaluate generalization performance, we test Pro-Trans under cross-dataset, cross-threat, and cross-attack settings. As shown in Table 5, traditional AP methods suffer from poor cross-dataset generalization, while Pro-Trans, relying only on intrinsic image properties (low-rankness and smoothness), exhibits greater flexibility. In Table 6, $l_\infty$ and $l_2$ indicate the threat model used during adversarial training. Table 6 shows that AP methods outperform AT under cross-threat setting, and Pro-Trans achieves the best robustness, improving RA by 1.0% and 4.2% over TNP under $l_\infty$ and $l_2$, respectively. In the cross-attack setting, as show in Table 7, Pro-Trans still delivers the best overall performance, with average RA gains of 5.37%.

Table 5: Standard and robust accuracy (%) against AutoAttack $l_\infty$ threat ($\epsilon = 8/255$) on CIFAR-10 and CIFAR-100 with WideResNet-28-10 classifier.

| Defense method | CIFAR-10 | | CIFAR-100 | | Average | |
|---|---|---|---|---|---|---|
| | SA | RA | SA | RA | SA | RA |
| AT (Cui et al., 2024) | 91.99 | 68.55 | 72.85 | 39.45 | 82.42 | 54.00 |
| AP (Nie et al., 2022) | 89.02 | 70.64 | 38.09 | 33.79 | 63.56 | 52.22 |
| TNP (Lin et al., 2025)* | 91.99 | 72.85 | 62.30 | 44.34 | 77.14 | 58.59 |
| Ours* | 87.69 | 74.02 | 65.62 | 45.11 | 76.65 | 59.56 |

## 4.4 Convergence Analysis

To evaluate PTR's convergence performance, we compare it with PuTT, QTT, and QTR on CIFAR-10 and ImageNet. During optimization, we record the loss at each iteration and report final reconstruction metrics including Peak Signal-to-Noise Ratio(PSNR), Normalized Root Mean Square Error(NRMSE), and Mean Squared Error(MSE).

As shown in Figure 4, PTR demonstrates faster convergence and better convergence performance. PTR converged in only 937 and 1561 iterations on CIFAR-10 and ImageNet, respectively, with a

Table 6: Standard and robust accuracy (%) against AutoAttack $l_\infty$ ($\epsilon = 8/255$) and $l_2$ ($\epsilon = 1.0$) threats on CIFAR-10 with standard ResNet-50 classifier.

| Type | Defense method | SA | RA | |
|---|---|---|---|---|
| | | | $l_\infty$ | $l_2$ |
| AT | Engstrom et al. (2019) $l_\infty$ | 89.8 | 52.1 | 27.7 |
| | Engstrom et al. (2019) $l_2$ | 92.1 | 30.6 | 38.0 |
| | Chen et al. (2020) $l_\infty$ | 87.7 | 52.1 | 32.4 |
| | Augustin et al. (2020)$^\dagger$ $l_2$ | 91.8 | 41.6 | 47.2 |
| AP | Nie et al. (2022) | 88.2 | 70.0 | 70.9 |
| | Lin et al. (2025) | 88.3 | 73.2 | 67.0 |
| | Ours | 86.3 | 74.2 | 71.2 |

Table 7: Standard and robust accuracy (%) against AutoAttack $l_\infty$ ($\epsilon = 8/255$) and PGD+EOT $l_\infty$ ($\epsilon = 8/255$) threats on CIFAR-10 with WideResNet-28-10 classifier.

| Type | Defense method | SA | RA | | |
|---|---|---|---|---|---|
| | | | PGD+EOT | AA | Avg. |
| AT | Gowal et al. (2020)$^\checkmark$ | 90.82 | 62.50 | 60.55 | 61.52 |
| | Rebuffi et al. (2021)$^\dagger$ | 88.48 | 64.26 | 60.35 | 62.30 |
| | Gowal et al. (2021)$^\dagger$ | 89.06 | 65.04 | 63.28 | 64.16 |
| AP | Yoon et al. (2021) | 86.76 | 37.11 | 60.86 | 48.99 |
| | Nie et al. (2022) | 90.43 | 51.13 | 66.06 | 58.60 |
| | Lee & Kim (2023) | 90.53 | 56.88 | 70.31 | 63.60 |
| | Ours* | 88.28 | 66.60 | 72.46 | 69.53 |

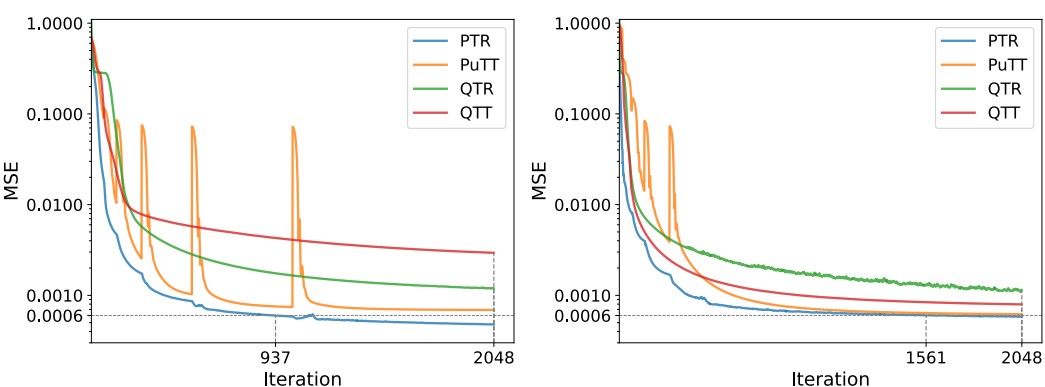

Figure 4: Loss curves comparison on CIFAR-10 (left) and ImageNet (right).

final reconstruction loss lower than that of the other methods. As novel coarse-to-fine TNs, PTR and PuTT both demonstrate higher efficiency compared to traditional QTT and QTR. However, PuTT requires interpolation-based upsampling via MPO followed by TT-SVD to compress redundant ranks, which leads to unstable reconstruction performance. It is observed that the PuTT optimization process exhibits noticeable jumps, which severely impact its convergence performance. In contrast, PTR maintains a stable decrease in reconstruction loss, resulting in a more efficient optimization process. Moreover, as shown in Table 8 and Table 9, PTR achieved the best reconstruction performance with the fewest parameters, consistently outperforming other TNs in PSNR, NRMSE, and MSE across datasets.

Table 8: Reconstruction performance comparison on CIFAR-10.

| TN | Parameters | PSNR | NRMSE | MSE |
|---|---|---|---|---|
| QTT | 27420 | 32.11 | 0.0550 | 0.00294 |
| QTR | 26955 | 30.28 | 0.0666 | 0.00119 |
| PuTT | 27025 | 33.41 | 0.0477 | 0.00069 |
| PTR | 26955 | 34.02 | 0.0463 | 0.00047 |

Table 9: Reconstruction performance comparison on ImageNet.

| TN | Parameters | PSNR | NRMSE | MSE |
|---|---|---|---|---|
| QTT | 2296 | 25.76 | 0.1036 | 0.00079 |
| QTR | 2187 | 29.74 | 0.0668 | 0.00114 |
| PuTT | 2237 | 32.36 | 0.0502 | 0.00061 |
| PTR | 2187 | 33.70 | 0.0423 | 0.00057 |

## 4.5 VISUALIZATION

To highlight the effect of AGLSR, we visualize the reconstructions of adversarial examples (AE) by PTR, Pro-Trans (w/o attention mask), and Pro-Trans in Figure 5, alongside the clean example (CE) and ground-truth perturbation. PTR achieves small reconstruction errors but also restores perturbations. Pro-Trans (w/o attention mask) suppresses perturbations more effectively, though with noticeable blurring. By applying attention mask, Pro-Trans better preserves semantic details while removing perturbations, achieving a more balanced purification effect.

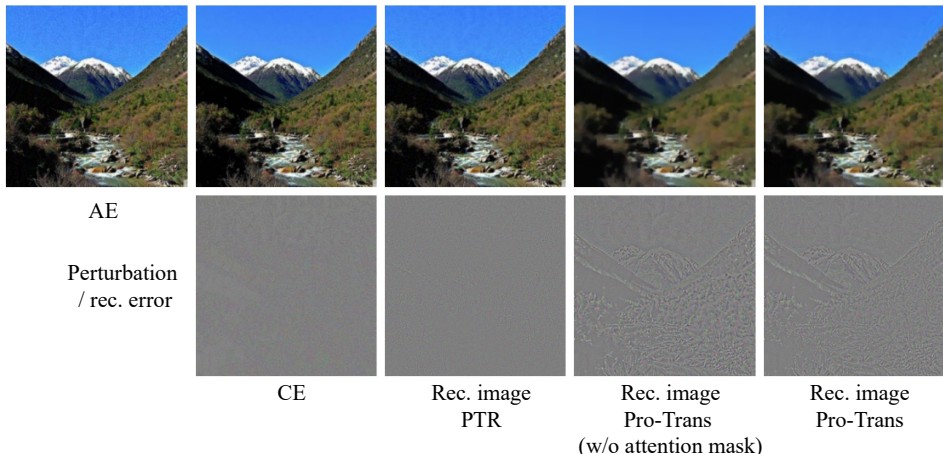

Figure 5: Visualization comparison of purification performance. The first row shows the adversarial example, clean example, and reconstructed images from PTR, Pro-Trans (w/o attention mask), and Pro-Trans. The second row presents the corresponding perturbations or reconstruction errors.

To provide a more intuitive demonstration of the attention mask, we visualize one image together with four activation maps generated by the ResNet-50 classifier and the corresponding attention mask, as shown in Figure 10. It can be clearly seen that the model pays stronger attention to decision-critical regions while assigning lower attention to the background. Therefore, the attention mask guides PTR to apply stronger local smoothing in critical regions while preserving more details in other areas. These results further highlight the effectiveness of our method.

## 4.6 ABLATION STUDY

Table 10: Standard and robust accuracy (%) against AutoAttack $l_\infty$ ($\epsilon = 8/255$) threat and average purification time (s) on ImageNet.

| Method | Time | SA | RA |
|---|---|---|---|
| TNP (Lin et al., 2025) | 8.002 | 65.43 | 42.77 |
| PTR | 2.054 | 66.01 | 47.07 |
| Pro-Trans (w/o attention mask) | 3.093 | 62.10 | 51.36 |
| Pro-Trans | 2.997 | 64.84 | 51.17 |

We conduct ablation studies on ImageNet, comparing TNP, PTR, Pro-Trans without attention mask, and Pro-Trans (Table 10). Compared to TNP, all PTR-based methods greatly reduce purification time (from 8s to 2–3s). Introducing local smoothing regularization (Pro-Trans w/o attention mask) substantially improves RA over both TNP and PTR, but at the cost of reduced SA, indicating that semantic details are over-smoothed. By incorporating the attention mask, Pro-Trans achieves nearly the same RA as Pro-Trans w/o attention mask but recovers much of the SA, thereby mitigating semantic degradation and achieving the most favorable RA–SA trade-off.

## 5 CONCLUSION

In this work, we introduced Pro-Trans, a novel tensor-based AP method that integrates PTR with AGLSR. PTR avoids redundant upsampling operations and enables coarse-to-fine optimization, significantly improving convergence efficiency and stability. Meanwhile, AGLSR leverages feature-level attention to adaptively apply local smoothing regularization, effectively suppressing perturbation while retaining semantic fidelity. Experiments on CIFAR-10, CIFAR-100, and ImageNet demonstrated that Pro-Trans achieves state-of-the-art robustness and strong generalization on cross-dataset, cross-threat, and cross-attack scenarios, with significantly reduced computational overhead, thus offering a favorable balance between robustness, efficiency, and generalization. These results establish TNs as a promising method for practical and generalizable adversarial purification.

**Limitations:** Despite the large improvements achieved and extensive experiments empirically supporting our claims, our work currently lacks a comprehensive theoretical analysis to fully explain the observed robustness. We leave this as an important direction for future research.

## ETHICS STATEMENT

This work only relies on publicly available datasets (CIFAR-10, CIFAR-100, ImageNet) and does not involve human subjects, private, or sensitive data. Our method is designed to improve the robustness and security of machine learning models against adversarial attacks. It does not introduce direct negative societal impacts, as its primary purpose is to strengthen defenses rather than enable new forms of attack.

## REPRODUCIBILITY STATEMENT

We have taken care to ensure the reproducibility of our work. All experimental settings, including optimization details, hyperparameters, and initialization strategies, are described in the main text and Appendix. The datasets used are all publicly available. The source code and scripts for reproducing our results will be released upon paper acceptance.

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

# A APPENDIX

## A.1 LLM USAGE DISCLOSURE

We used large language models (LLMs) to assist with improving the clarity, grammar, and phrasing of the manuscript. No LLMs were used to generate novel method content, and all technical contributions were developed and validated solely by the authors.

## A.2 IDENTITY TENSOR

In PTR, each core tensor models information at a specific granularity. The two dimensions that connect to other granularities capture inter-granularity relationships, while the remaining physical dimension represents the information of the current granularity. Core tensors that are not optimized at a given stage are initialized as identity tensors, which serve to replicate pixels in the reconstruction process. This design corresponds to the initialization of PTR's optimization objective, where images of different resolutions are upsampled to the original resolution through pixel replication. Concretely, we first construct a diagonal matrix and then stack it along the physical mode to obtain the identity tensor, formally expressed as follows:

$$\mathsf{I} \in \mathbb{R}^{rank \times 4 \times rank}, \quad \mathsf{I}_{i,j,k} = \delta_{ik},$$

where $\delta_{ik}$ denotes the Kronecker delta, i.e.,

$$\delta_{ik} = \begin{cases} 1 & \text{if } i = k, \\ 0 & \text{otherwise.} \end{cases}$$

Thus, for any fixed $j$, the slice $\mathcal{I}_{:j:}$ corresponds to an $rank \times rank$ identity matrix. Figure 6 provides a more intuitive visualization of the identity tensor. In practice, this construction ensures that non-optimized core tensors act as pixel replication operators during the coarse-to-fine reconstruction process in PTR.

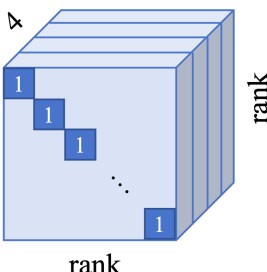

Figure 6: Illustration of Identity Tensor.

## A.3 IMPLEMENTATION DETAILS

We provide here a comprehensive description of the experimental settings and hyperparameters to ensure clarity and reproducibility of our results. For all TN–based experiments, we used the Adam optimizer with a learning rate of 0.008 and optimized for 2048 iterations. Images from CIFAR-10 and CIFAR-100 ($32 \times 32$) were upsampled via bilinear interpolation to $256 \times 256$, while ImageNet images ($224 \times 224$) were upsampled to $256 \times 256$.

### DEFENSE EVALUATION

For adversarial defense experiments, we set the maximum rank of TNP to 10 and that of Pro-Trans to 14 on CIFAR-10/100. For ImageNet, the maximum rank of TNP was set to 200 and that of Pro-Trans to 50.

CONVERGENCE ANALYSIS

For PuTT and PTR, the initial optimization resolution was $8 \times 8$. On CIFAR-10, the optimization target switched progressively at steps 64, 128, 256, 512, and 1024, while on ImageNet the switching steps were 16, 32, 64, 128, and 256. On CIFAR-10, the maximum rank was set to 10 for Tensor Train-based QTT and PuTT, and 9 for Tensor Ring-based QTR and PTR. On ImageNet, the maximum rank was set to 43 for QTT, 100 for PuTT, and 33 for TR-based QTR and PTR.

Table 11: Summary of experimental settings and hyperparameters.

| Setting | CIFAR-10/100 | ImageNet |
|---|---|---|
| Optimizer | Adam (lr = 0.008, 2048 iters) | Adam (lr = 0.008, 2048 iters) |
| Image resolution | $32 \times 32 \to 256 \times 256$ | $224 \times 224 \to 256 \times 256$ |
| **Defense evaluation** | | |
| TNP rank | 10 | 200 |
| Pro-Trans rank | 14 | 50 |
| **Convergence analysis** | | |
| Initial resolution | $8 \times 8$ | $8 \times 8$ |
| Switching steps | 64, 128, 256, 512, 1024 | 16, 32, 64, 128, 256 |
| QTT rank | 10 | 43 |
| PuTT rank | 10 | 100 |
| QTR rank | 9 | 33 |
| PTR rank | 9 | 33 |

## A.4 ADDITIONAL VISUALIZATIONS

In this section, we present additional visualization results of ImageNet for reference, including reconstruction outputs across different resolutions, extended comparative visualizations, and attention mask visualizations.

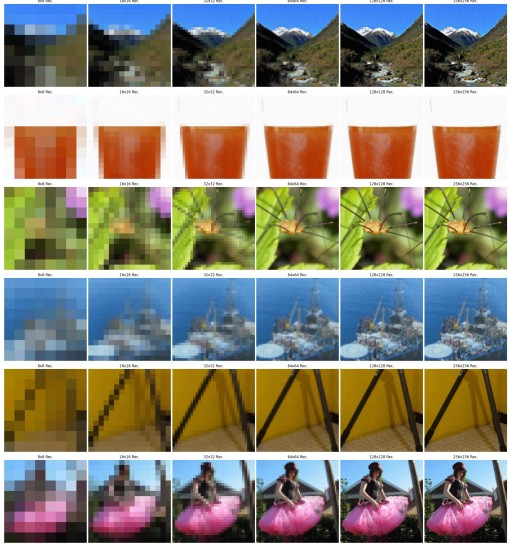

Figure 7: PTR reconstruction results at different resolutions.

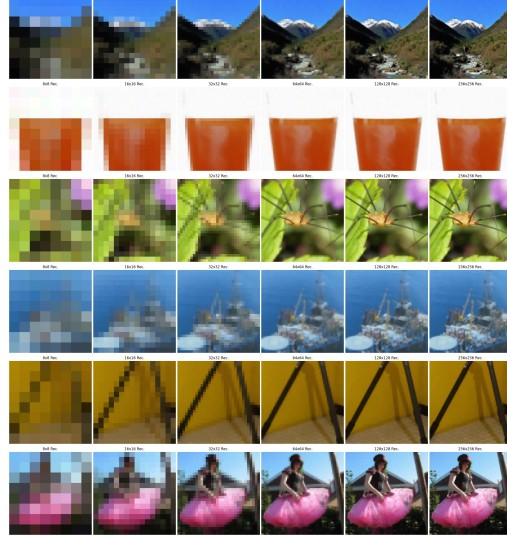

Figure 8: Pro-Trans reconstruction results at different resolutions.

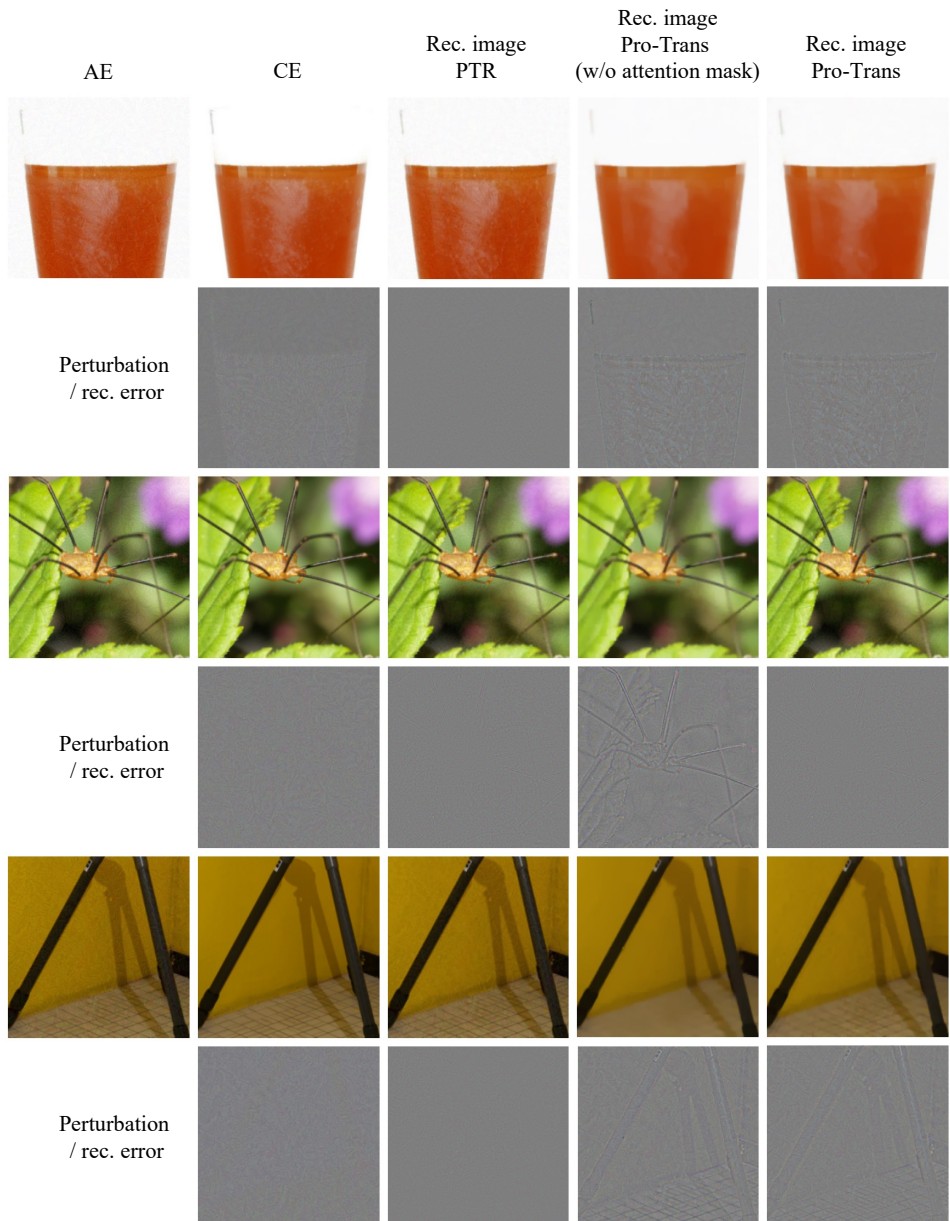

Figure 9: More Visualization Comparison of Purification Effect.

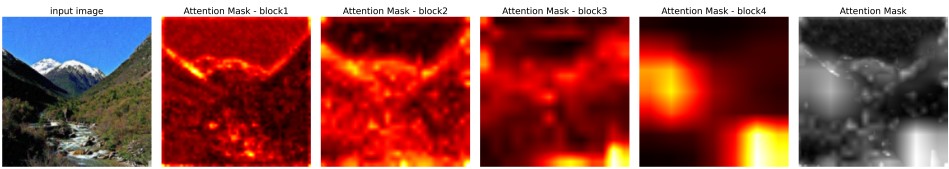

Figure 10: Visualization of the attention mask. From left to right are the input image, its activation maps from ResNet-50, and the derived attention mask.

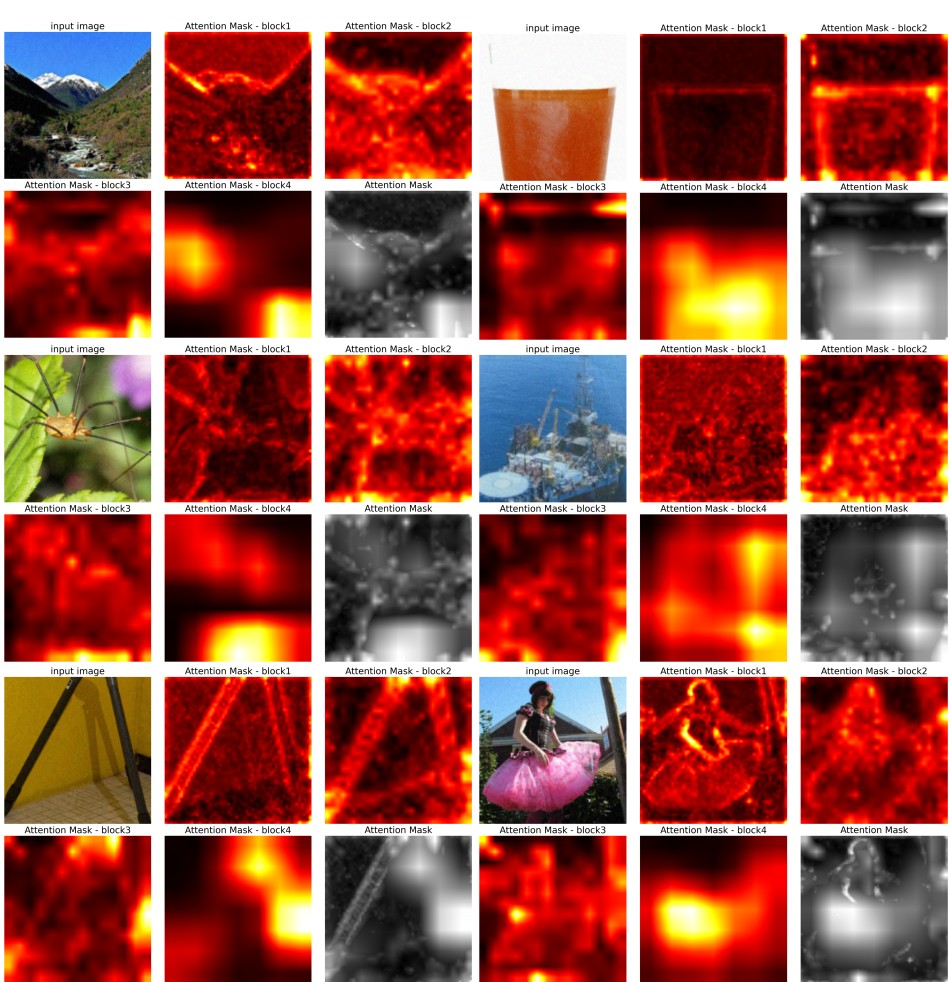

Figure 11: Visualization of Attention Mask.

