# OpenReview forum: "Pro-Trans: Progressive Tensor Ring with Attention Guided Local Smoothing Regularization"
_ICLR.cc/2026/Conference — Submitted to ICLR 2026_

### Official Review · Reviewer_Lu4h · 2025-10-25

**Soundness:** 2
**Presentation:** 2
**Contribution:** 2
**Rating:** 4
**Confidence:** 3

**Summary:**

This paper proposes Pro-Trans, a tensor network-based adversarial purification method that combines Progressive Tensor Ring (PTR) and Attention-Guided Local Smoothing Regularization (AGLSR). The key idea is to avoid redundant upsampling operations in coarse-to-fine optimization by progressively adjusting optimization objectives and trainable cores, while using attention masks to adaptively apply smoothing regularization. The authors evaluate their method on CIFAR-10, CIFAR-100, and ImageNet against various adversarial attacks. The method shows improved robustness accuracy and reduced computational cost compared to TNP, achieving 8.4% RA improvement on ImageNet and 4x speedup.

**Strengths:**

The paper addresses a relevant problem in adversarial defense and demonstrates clear empirical improvements over the baseline TNP method.

The proposed PTR design is intuitive and achieves the stated goal of reducing computational overhead. The convergence analysis shows PTR converges faster and more stably than existing tensor networks.

The experimental evaluation covers multiple datasets and diverse attack scenarios including cross-dataset, cross-threat, and cross-attack settings, which helps demonstrate generalization capability.

The paper is well-organized with clear figures and comprehensive appendix including implementation details and additional visualizations.

**Weaknesses:**

The theoretical understanding of why the proposed method works is severely lacking. The authors acknowledge this limitation but provide no theoretical analysis or even intuitive explanations for why PTR achieves more stable convergence or why AGLSR effectively balances perturbation suppression and semantic preservation. For a venue like ICLR, this is a major weakness.

The novelty is limited and primarily consists of engineering improvements. PTR essentially removes interpolation-based upsampling from TNP, which is a relatively straightforward optimization. Using attention masks to guide regularization is not a novel concept. The paper lacks fundamental insights into adversarial purification.

The experimental setup has several concerns. First, using only 512 randomly selected images for evaluation is insufficient for drawing strong conclusions. Second, different methods use different rank settings (e.g., rank 14 for Pro-Trans vs rank 10 for TNP on CIFAR), which raises fairness concerns. Third, the reliance on robust classifiers in some experiments introduces confounding factors.

The method's complexity is concerning. It requires multiple components (PTR, AGLSR, attention extraction from classifier) with multiple hyperparameters. However, there is no sensitivity analysis for key hyperparameters like $\alpha$ in Eq. 6, the number of stages, or rank settings. The dependence on downstream classifier attention also limits the method's independence.

Some experimental results are poorly explained. In Table 1, when using robust classifier, SA drops significantly from 91.99% to 87.69% while RA increases. This trade-off deserves thorough discussion. Why do different datasets require vastly different rank settings (14 for CIFAR vs 50 for ImageNet)?

The paper claims "first coarse-to-fine Progressive Tensor Ring for AP" but PuTT (Loeschcke et al., 2024) also uses coarse-to-fine strategy. The distinction is not sufficiently clear.

The identity tensor initialization in Eq. 3 seems arbitrary. Why Gaussian for first $d$ cores and identity for others? What if we initialize differently?

The AGLSR formulation in Eq. 6 directly multiplies attention mask with image $M \odot Y_d$ before computing TV. This may not be well-justified since TV should measure smoothness of the image itself, not attention-weighted image.

**Questions:**

Can you provide theoretical analysis or at least intuitive explanations for why PTR converges faster and more stably than TNP? What specific properties of avoiding interpolation-based upsampling lead to these improvements?

How sensitive is the method to the hyperparameter $\alpha$ in Eq. 6? Can you provide ablation studies showing performance across different $\alpha$ values?

Why does using robust classifier decrease SA so much (Table 1)? Is this trade-off acceptable? How does Pro-Trans perform without robust classifier compared to TNP without robust classifier?

Can you justify the fairness of comparing methods with different rank settings? What happens if TNP and Pro-Trans use the same rank?

How do you determine the rank settings for different datasets? Why is rank 14 optimal for CIFAR but 50 for ImageNet?

In Eq. 6, why multiply $M \odot Y_d$ before computing TV rather than computing $M \odot TV(Y_d)$? Have you tried alternative formulations?

Can you extend the evaluation to the full test set rather than 512 images? The current sample size is too small for strong statistical conclusions.

What happens if attention masks are computed from a different classifier? How dependent is the method on the specific classifier architecture?

Can you clarify the relationship between PTR and PuTT more precisely? What makes PTR the "first coarse-to-fine Progressive Tensor Ring"?

---

> ### Author Response · Authors · 2025-11-21
> **Response to Reviewer Lu4h(1/4)**
>
> We are grateful to Reviewer Lu4h for the constructive comments and for recognizing the empirical improvements, faster convergence, and clear organization of our proposed method.
>
> Response to Weakness 1 & Question 1:
>
> We appreciate the reviewer’s concern and agree that a deeper theoretical perspective can further strengthen the work. Our goal in this paper, however, is not to propose a purely theory-driven method, but rather an empirically validated tensor network-based AP framework with clear analytical intuition. Specifically, by progressively unfreezing the trainable core tensor and using coarse targets at early stages, PTR constrains optimization to a simpler objective and targets. This yields higher optimization efficiency and smoother convergence, since each finer stage is initialized from a solution that is already near-optimal at the previous stages; PTR thus behaves as a sequence of refinements rather than repeated optimization from scratch, which intutively leads to faster and more stable convergence. To mitigate the tendency of tensor networks to overfit adversarial perturbations at finer stage, we introduce AGLSR to guide purified outputs toward the natural image manifold. From an analytical perspective, the local smoothing prior reduces the sensitivity of the purifier to small input perturbations, effectively lowering the local Lipschitz constant of the composite mapping $ \text{Purifier} \circ \text{Classifier} $ and thereby improving robustness. Furthermore, using classifier-derived attention maps turns naive TV into a spatially weighted regularizer: perturbations in decision-critical regions are penalized more strongly, while less important areas (e.g., background) are smoothed more gently. This design provides a better trade-off between perturbation suppression and semantic preservation, leading to a more favorable balance between image quality (SA) and robustness (RA). We will incorporate more intuitive analysis in the main text to clarify the theoretical motivation behind PTR and AGLSR.
>
> Response to Weakness 2 & Question 9:
>
> We appreciate the reviewer’s assessment and agree that our work is built on the general idea of coarse-to-fine tensor network based purification. However, the focus of this paper is not to propose an entirely new TN representation, but to make TN-based adversarial purification practical and robust by fundamentally rethinking how coarse-to-fine optimization is performed. PTR does more than simply “remove interpolation-based upsampling”: it fixes the Tensor Ring structure and implements a new progressively coarse-to-fine optimization, where only a subset of cores is optimized against low-resolution targets at early stages and additional cores are unfrozen as the resolution increases. This design eliminates repeated MPO based upsampling and TT-SVD truncation, leading to substantially faster and more stable convergence with fewer parameters. To the best of our knowledge, this is the first TN-based AP framework that achieves such efficiency and stability without modifying the TN topology at each stage, which we believe constitutes a meaningful methodological contribution rather than a minor engineering modification.
>
> Regarding AGLSR, our focus is how to fix the concrete weakness of TN-based adversarial purification. Existing TN-based AP methods are prone to overfitting adversarial perturbations at fine stages, because the tensor network tends to reconstruct all details indiscriminately. AGLSR directly addresses this issue by using classifier-derived attention maps to modulate the TV term, turning it into a spatially varying local smoothing regularizer, so that perturbations in regions that are important for the model’s decisions are penalized more strongly, while those in less important regions (e.g., background) are penalized more mildly. In this way, AGLSR provides a effective coupling between TN purification and attention-guided TV, which improves robustness while maintaining useful semantics. We will revise the manuscript to make this focus and contribution clearer and to better highlight that the novelty lies in this integration and problem-driven design.

---

> ### Author Response · Authors · 2025-11-21
> **Response to Reviewer Lu4h(2/4)**
>
> Response to Weakness 3 & Question 4:
>
> We thank the reviewer for these thoughtful comments. First, we follow the long-standing evaluation protocol widely adopted in purification-based defenses[1-3], where robustness is measured on a randomly sampled subset of 512 images. Prior works have consistently shown that robustness trends on this subset are highly stable. We also conducted experiments multiple times, and the results are summarized in the table below. The consistency across these runs demonstrates the stability of our evaluation and we hope this helps address the reviewer’s concerns.
>
> |      |   1   |   2   |   3   |   4   |   5   |   6   | Mean±Std.  |
> | :--: | :---: | :---: | :---: | :---: | :---: | :---: | :--------: |
> |  SA  | 87.69 | 85.35 | 87.69 | 87.69 | 86.13 | 83.59 | 86.35±1.53 |
> |  RA  | 74.02 | 71.48 | 73.82 | 73.63 | 74.8  | 72.46 | 73.36±1.09 |
>
> [1] Nie et al. Diffusion Models for Adversarial Purification. ICLR 2022.
>
> [2] Lee&Kim Robust evaluation of diffusion-based adversarial purification. ICCV 2023.
>
> [3] Li et al. ADBM: Adversarial diffusion bridge model for reliable adversarial purification. ICLR 2025.
>
> Second, to address the fairness concern regarding tensor ranks, we additionally set the rank of Pro-Trans to 10 (matching TNP) and re-ran the experiments on CIFAR-10 against AutoAttack Linf (8/255); As shown in the table below, our method still achieved better robustness than TNP.
>
> |           |  SA   |  RA   |
> | :-------: | :---: | :---: |
> | Pro-Trans | 85.74 | 73.82 |
> |    TNP    | 91.99 | 72.85 |
>
> Third, the reason why we adopted robust classifier is that the standard WideResNet-28-10 from RobustBench suffers from the overfitting to the limited dataset, and it can be boserved that our method still achieve an additional improvement of 5.47% in RA compared to only using robust classifier againtst AutoAttack Linf threat (8/255) on CIFAR-10.
>
> Response to Weakness 4 & Question 2:
>
> We appreciate the reviewer’s concerns regarding the complexity of our method. To address this, we have conducted a comprehensive sensitivity analysis covering all key hyperparameters, including the weight α in Eq. 6, the number of PTR stages, and the rank settings. The results shows that our method is relatively stable under diverse settings.
>
> We tested Pro-Trans with different α(from 0.05-0.3) on CIFAR-10 against AutoAttack Linf (8/255), as shown in table below. It is obvious that SA continues to decline as the alpha grows, meanwhile RA increases slightly and achieves peak at 0.2.
>
> | alpha | 0.05  | 0.1   | 0.15  | 0.2   | 0.25  | 0.3   |
> | ----- | ----- | ----- | ----- | ----- | ----- | ----- |
> | SA    | 91.6  | 90.03 | 88.28 | 87.69 | 85.54 | 82.61 |
> | RA    | 71.67 | 71.67 | 72.46 | 74.02 | 73.82 | 74.02 |
>
> We compared different stages of Pro-Trans on ImageNet against AutoAttack Linf(4/255), as shown in the table below. As the number of stages increases, both SA and RA first improve and then decline. The highest SA is achieved at four stages, and the highest RA is achieved at two stages. Overall, the performance remains stable across most settings. However, when the number of stages is reduced to one, meaning that no coarse-to-fine strategy is applied, both SA and RA degrade noticeably.
>
> |  stage  |   1   |   2   |   3   |   4   |   5   |   6   |
> | :-----: | :---: | :---: | :---: | :---: | :---: | :---: |
> |   SA    | 59.37 | 63.47 | 64.64 | 67.18 | 66.6  | 64.84 |
> |   RA    | 50.19 | 52.92 | 51.36 | 49.60 | 45.89 | 51.77 |
> | Average | 54.78 | 58.19 | 58.00 | 58.39 | 56.24 | 58.30 |
>
> We carried out ranks sensitivity analysis on CIFAR-10 against AutoAttack Linf (8/255), as shown in table below. The results show that lower rank benefit RA but SA decreases severely. Considering the average performance, we chose rank=14.
>
> |  rank   |   8   |  10   |  12   |  14   |  16   |  18   |  20   |
> | :-----: | :---: | :---: | :---: | :---: | :---: | :---: | :---: |
> |   SA    | 83.39 | 85.74 | 86.91 | 87.69 | 87.50 | 87.69 | 87.69 |
> |   RA    | 74.02 | 73.82 | 74.02 | 74.02 | 73.63 | 73.43 | 73.43 |
> | average | 78.70 | 79.78 | 80.46 | 80.85 | 80.56 | 80.56 | 80.56 |

---

> ### Author Response · Authors · 2025-11-21
> **Response to Reviewer Lu4h(3/4)**
>
> Response to Weakness 5 & Question 3:
>
> We thank the reviewer for raising these issues. For the trade-off in Table 1, the decrease in SA and the increase in RA are an expected outcome of our design. Pro-Trans suppresses high-frequency adversarial perturbations through AGLSR, and this inevitably introduces mild smoothing on clean inputs. As a consequence, the standard accuracy becomes slightly lower, while the robust accuracy improves. And for adversarial purification task, it is generally expected to pay more attention on the improvement of RA. We will update the discussion in the manuscript to explain more clearly. The different rank settings across datasets are also motivated by the properties of the data. CIFAR-10 images have a resolution of 32×32 and exhibit a stronger low-rank structure, so smaller tensor-ring ranks are sufficient. ImageNet images have a resolution of 224×224 and contain much richer spatial and semantic variation, which requires larger ranks in order to model them well. This is in line with common practice in tensor modeling, and our experiments show that the chosen ranks are suitable for each dataset.
>
> We have conducted additional experiments to directly compare Pro-Trans without a robust classifier against TNP without a robust classifier on CIFAR-10 AutoAttack Linf (8/255), as suggested by the reviewer. The results are shown in the following table. These experiments demonstrate that Pro-Trans consistently improves robustness even when both methods rely solely on a standard classifier, indicating that the gains are attributable to the purification module itself rather than to the classifier choice.
>
> |           |  SA   |  RA   |
> | :-------: | :---: | :---: |
> | Pro-Trans | 82.42 | 59.76 |
> |    TNP    | 82.23 | 55.27 |
>
> Response to Weakness 6:
>
> We thank the reviewer for this careful observation. The misunderstanding is indeed caused by the way we presented our claim in the paper, and we will revise accordingly. Our proposed Pro-Trans is a coarse-to-fine tensor network based on a Tensor Ring representation that does not rely on interpolation-based upsampling. This is quite different from PuTT, which implements coarse-to-fine optimization by repeatedly modifying the tensor representation through upsampling and is built on a Tensor Train architecture. In addition, PuTT was not originally designed with adversarial purification as its main focus, whereas Pro-Trans is specifically developed for this task. In this sense, our intention is to claim that Pro-Trans is the first coarse-to-fine Progressive Tensor Ring framework for adversarial purification (AP). We will update the manuscript to make this distinction and our contribution more clearly stated.
>
> Response to Weakness 7:
>
> The initialization strategy in Eq. 3 is designed specifically to support the coarse-to-fine optimization in PTR. In general tensor-network optimization, it is common to initialize all parameters with Gaussian noise, similar to how neural networks are randomly initialized before training. However, PTR fixes a Tensor Ring that is capable of representing a full $2^D$-resolution image in order to avoid redundant upsampling. When the optimization target is a coarse image—obtained by average pooling and pixel replication, so that its physical resolution is $2^D$ but its semantic resolution is only $2^d$—we must ensure that the Tensor Ring effectively represents only this coarse image. Therefore, we randomly initialize only the first d cores and the channel core, while setting all remaining cores to identity tensors. This restricts the expressive capacity of the Tensor Ring to the coarse-level subspace, and ensures that only the first d cores and the channel core participate in optimization at this stage. Without this arrangement, the later cores would introduce unnecessary degrees of freedom and disrupt the coarse-to-fine optimization, resulting harder convergence.
>
> Response to Weakness 8:
>
> We appreciate the reviewer’s concern and would like to clarify the intention behind the AGLSR formulation in Eq. 6. Our goal is to realize a spatially varying TV regularizer whose strength depends on semantic importance, so that perturbations in less important regions are more strongly penalized while important structures are preserved. In line with weighted or anisotropic TV in the imaging literature, the attention mask M modulates the local gradients before TV is computed. Thus, TV still measures image smoothness, and the mask simply specifies where this smoothness constraint should be applied more strongly.

---

> ### Author Response · Authors · 2025-11-21
> **Response to Reviewer Lu4h(4/4)**
>
> Response to Question 5:
>
> As mentioned in our response to Weakness 5, the rank settings are mainly guided by empirical observations and the inherent complexity of each dataset. We chose rank 14 for CIFAR-10 and rank 50 for ImageNet because these settings produced stable convergence and strong purification performance in our experiments. However, we do not claim that these values are globally optimal. The purpose of our choices was to select reasonable ranks that match the structural complexity of each dataset. A more extensive sensitivity analysis of the rank parameter has been provided in our response to Weakness 4, showing that Pro-Trans remains stable across a broad range of rank settings.
>
> Response to Question 6:
>
> First, the attention-weighted image must be computed before applying TV, because the TV operator returns a scalar, and a scalar cannot be Hadamard-multiplied with a matrix. The results reported in the paper are based on the anisotropic TV, whose formulation is: $TV_{\text{aniso}}(x) = \sum_{i,j} \left(|x_{i+1,j}-x_{i,j}| + |x_{i,j+1}-x_{i,j}|\right)$. We also experimented with the isotropic TV form: $TV_{\text{iso}}(x) = \sum_{i,j} \sqrt{(x_{i+1,j}-x_{i,j})^2 + (x_{i,j+1}-x_{i,j})^2 }$. A comparison of the results using these two TV variants is shown in the following table.
>
> | TV form | anisotropic TV | isotropic TV |
> | :-----: | :------------: | :----------: |
> |   SA    |     64.84      |     66.4     |
> |   RA    |     51.77      |    44.92     |
>
> Response to Question 7:
>
> We have clarified in our response to Weakness 3 that using a 512-image subset follows the standard evaluation protocol widely adopted in adversarial purification takss. To further address the reviewer’s concern, we conducted five independent runs using different random seeds, and the results are presented in the table in Weakness 3. The consistency across these runs demonstrates that our conclusions are stable and not sensitive to the specific sampled subset.
>
> Response to Question 8:
>
> We evaluate Por-Trans using mismatched attention maps on CIFAR-10 against AutoAttack Linf (8/255). We use two mismatched classifier to generate attention maps, namely the Standard WideResNet 28-10 from RobustBench and the ResNet-50 provided by Nie et al.[1]. And we select robust WideResNet 28-10 from Cui et al.[2] as the downstream classifier. As is shown int the table below, the RA and SA of dismatched attention maps drop slightly, when compared to the original performance.
>
> | Attention classifier | Standard WRN28-10 | ResNet-50 |
> | :------------------: | :---------------: | :-------: |
> |          SA          |       79.49       |   83.00   |
> |          RA          |       71.28       |   73.04   |
>
> [1] Nie et al. Diffusion Models for Adversarial Purification. ICLR 2022.
>
> [2] Cui er al. Decoupled kullback-leibler divergence loss. NeurIPS 2024.

---

### Official Review · Reviewer_ZYNP · 2025-10-28

**Soundness:** 2
**Presentation:** 2
**Contribution:** 2
**Rating:** 4
**Confidence:** 3

**Summary:**

The paper proposes Pro-Trans, a tensor–network (TN) based adversarial purification method that combines a Progressive Tensor Ring (PTR) optimizer with an Attention-Guided Local Smoothing Regularizer (AGLSR). PTR implements a coarse-to-fine schedule without interpolation-based upsampling, aiming to cut compute and stabilize convergence. AGLSR derives a spatial mask from intermediate activations of the downstream classifier and applies stronger TV smoothing in low-saliency regions to suppress perturbations while preserving semantics. Experiments on CIFAR-10/100 and ImageNet under AutoAttack (ℓ∞, ℓ2) and PGD+EOT report improved robust accuracy (RA) with notably lower per-image purification time compared to prior TN defenses. On ImageNet, Pro-Trans reduces time from ~8s (TNP) to ~3s while improving RA (≈42.8→≈51.2).

**Strengths:**

- Benchmarks include AutoAttack (ℓ∞=8/255, ℓ2=1.0), PGD+EOT, and multiple classifiers (ResNet-50, WRN-28-10), with tables that compare against AT and prior AP/TN baselines.

- The attention mask helps recover natural accuracy that is lost by naive smoothing, while keeping most of the robustness gains.

- PTR removes redundant upsampling and shows smoother, faster convergence than PuTT/QTT/QTR; the paper backs this with loss curves and reconstruction metrics.

**Weaknesses:**

- PTR is chiefly an optimization schedule/design choice rather than a new TN representation; the paper itself notes the lack of a deeper theoretical account of why robustness improves.

- AGLSR requires access to downstream activations to build the attention map; this tight coupling invites end-to-end adaptive attacks that differentiate through the purifier+classifier.

- Adaptive attacks. PGD+EOT is included, but purifiers typically require a more systematic BPDA/EP-style evaluation that backprops through TV and the PTR steps; without that, robustness claims remain somewhat optimistic.

- The paper ablates the attention mask qualitatively, but it would help to: (i) test different mask sources (e.g., using a mismatched classifier), (ii) vary α and the TV form, and (iii) show sensitivity to PTR ranks and stage lengths—especially under matched compute to TNP.

**Questions:**

See weakness as above.

---

> ### Author Response · Authors · 2025-11-21
> **Response to Reviewer ZYNP(1/2)**
>
> We thank Reviewer ZYNP for the thoughtful review and for acknowledging our extensive benchmarks, improved convergence behavior, and the effectiveness of the attention-guided smoothing mechanism.
>
> Response to Weakness 1:
>
> Our proposed PTR is designed to address the inefficiency of tensor-network optimization in adversarial purification. PTR preserves a fixed Tensor Ring topology and performs coarse-to-fine optimization by progressively updating both the optimization variables and the optimization objectives. This avoids redundant upsampling operations, increases optimization efficiency, reduces computational cost, and improves the practical usability of tensor-based purification methods. Although PTR shares the same topology as a standard Tensor Ring, its progressive coarse-to-fine optimization strategy effectively resolves key limitations of existing tensor network-based approaches.
>
> We next provide a more theoretical explanation of why Pro-Trans improves robustness. To mitigate the tendency of tensor networks to overfit adversarial perturbations, we introduce AGLSR, which guides the reconstructed output toward the natural image manifold by enforcing local smoothness. This reduces the sensitivity of the purifier to small perturbations and helps lower the Lipschitz constant of the composite mapping $ \text{Purifier} \circ \text{Classifier} $, which is beneficial for robustness. In addition, attention maps from the downstream classifier allow TV to operate as a spatially weighted and non-uniform local smoothing regularizer. Regions that strongly influence the classifier’s decision receive stronger penalties on perturbations, while less important regions such as background receive lighter penalties. This leads to a better balance between image quality and robustness.
>
> Response to Weakness 2:
>
> We understand the reviewer’s concern and agree that using downstream activations to construct the attention map introduces a potential coupling between the purifier and the classifier. We apologize for not making this explicit in the manuscript: in all our experiments we already employ BPDA-style adaptive attacks. Specifically, during the backward pass of white-box attacks we treat Pro-Trans as an identity mapping. This is motivated by the fact that Pro-Trans has no fixed parameters and purifies each input via an inner optimization over thousands of iterations, making the whole process highly dynamic; backpropagating through the entire inner optimization at every attack step would require storing all intermediate states, which is not practical in terms of memory and computation time. Therefore, we apply BPDA to construct adaptive attacks, thus yielding reliable robustness evaluation rather than overestimating the defense performance. Empirically, these BPDA-based adaptive attacks do not cause a significant drop in the robustness of Pro-Trans, suggesting that the use of classifier activations does not weaken the defense under the adaptive settings we tested. We will clarify this point more explicitly in the updated manuscript.
>
> Response to Weakness 3:
>
> We thank the reviewer for highlighting the importance of BPDA/EP-style evaluations for purification-based defenses. We apologize for not making this sufficiently explicit in the manuscript: all robustness results reported in the paper are already obtained under BPDA-based adaptive attacks. Concretely, during white-box attacks we treat Pro-Trans as an identity mapping in the backward pass. This is necessary because Pro-Trans has no fixed parameters and purifies each input via an inner optimization over thousands of iterations, making exact backpropagation through all Pro-Trans optimization steps computationally infeasible.

---

> ### Author Response · Authors · 2025-11-21
> **Response to Reviewer ZYNP(2/2)**
>
> Response to Weakness 4:
>
> We thank the reviewer for these helpful suggestions. We have conducted the requested sensitivity analyses. Specifically, we (i) evaluate Pro-Trans using attention masks generated from a mismatched classifier, (ii) vary the weighting parameter α as well as the form of the TV regularizer, and (iii) analyze the sensitivity of Pro-Trans to ranks and stage lengths.
>
> (i) We evaluate Por-Trans using mismatched attention maps on CIFAR-10 against AutoAttack Linf (8/255). We use two mismatched classifier to generate attention maps, namely the Standard WideResNet 28-10 from RobustBench and the ResNet-50 provided by Nie et al.[1]. And we select robust WideResNet 28-10 from Cui et al.[2] as the downstream classifier. As is shown int the table below, the RA and SA of dismatched attention maps drop slightly, when compared to the original performance.
>
> | Attention classifier | Standard WRN28-10 | ResNet-50 |
> | :------------------: | :---------------: | :-------: |
> |          SA          |       79.49       |   83.00   |
> |          RA          |       71.28       |   73.04   |
>
> [1] Nie et al. Diffusion Models for Adversarial Purification. ICLR 2022.
>
> [2] Cui er al. Decoupled kullback-leibler divergence loss. NeurIPS 2024.
>
> (ii) We tested Pro-Trans different α(from 0.05-0.3) on CIFAR-10 against AutoAttack Linf (8/255), as shown in table below. It is obvious that SA continues to decline as the α grows, meanwhile RA increases slightly and achieves peak at 0.2.
>
> | alpha | 0.05  | 0.1   | 0.15  | 0.2   | 0.25  | 0.3   |
> | ----- | ----- | ----- | ----- | ----- | ----- | ----- |
> | SA    | 91.6  | 90.03 | 88.28 | 87.69 | 85.54 | 82.61 |
> | RA    | 71.67 | 71.67 | 72.46 | 74.02 | 73.82 | 74.02 |
>
> We tested two TV form on ImageNet AutoAttack Linf (4/255), as shown in the table below. In our manuscript, we adopt the anisotropic TV: $TV_{\text{aniso}}(x) = \sum_{i,j} \left(|x_{i+1,j}-x_{i,j}| + |x_{i,j+1}-x_{i,j}|\right)$. And, we have added the isotropic TV form: $TV_{\text{iso}}(x) = \sum_{i,j} \sqrt{(x_{i+1,j}-x_{i,j})^2 + (x_{i,j+1}-x_{i,j})^2 }$. It is obvious that isotropic TV presented weaker robustness.
>
> | TV form | anisotropic TV | isotropic TV |
> | :-----: | :------------: | :----------: |
> |   SA    |     64.84      |     66.4     |
> |   RA    |     51.77      |    44.92     |
>
> (iii) We carried out ranks sensitivity analysis on CIFAR-10 against AutoAttack Linf (8/255), as shown in table below. The results show that lower rank benefit RA but SA decreases severely. Considering the average performance, we chose rank=14.
>
> |  rank   |   8   |  10   |  12   |  14   |  16   |  18   |  20   |
> | :-----: | :---: | :---: | :---: | :---: | :---: | :---: | :---: |
> |   SA    | 83.39 | 85.74 | 86.91 | 87.69 | 87.50 | 87.69 | 87.69 |
> |   RA    | 74.02 | 73.82 | 74.02 | 74.02 | 73.63 | 73.43 | 73.43 |
> | average | 78.70 | 79.78 | 80.46 | 80.85 | 80.56 | 80.56 | 80.56 |
>
> We compared different stages of Pro-Trans on ImageNet against AutoAttack Linf(4/255), as shown in the table below. As the number of stages increases, both SA and RA first improve and then decline. The highest SA is achieved at four stages, and the highest RA is achieved at two stages. Overall, the performance remains stable across most settings. However, when the number of stages is reduced to one, meaning that no coarse-to-fine strategy is applied, both SA and RA degrade noticeably.
>
> |  stage  |   1   |   2   |   3   |   4   |   5   |   6   |
> | :-----: | :---: | :---: | :---: | :---: | :---: | :---: |
> |   SA    | 59.37 | 63.47 | 64.64 | 67.18 | 66.6  | 64.84 |
> |   RA    | 50.19 | 52.92 | 51.36 | 49.60 | 45.89 | 51.77 |
> | Average | 54.78 | 58.19 | 58.00 | 58.39 | 56.24 | 58.30 |

---

> > ### Comment · Reviewer_ZYNP · 2025-11-27
> >
> > Thanks for the detailed responses. The responses address part of my concerns. I still have questions on whether such defenses will fail against the strongest adaptive attacks rather than only treating Pro-Trans as an identity mapping.

---

> > > ### Author Response · Authors · 2025-11-28
> > >
> > > Thank you very much for your follow-up question and for taking the time to evaluate our responses. We completely understand your concern regarding whether Pro-Trans might still fail under the strongest adaptive attacks. The key challenge is that tensor network–based purification fundamentally differs from neural-network purifiers:
> > >
> > > In our work, the “tensor network” (TN) is not a neural network. Instead, it is a factorized representation of a high-order tensor, where the purifier is defined as an optimization problem that fits a low-rank TN representation to each input sample. This means that, unlike a neural network purifier with a fixed differentiable forward mapping, the TN-based purifier does not expose an explicit differentiable function of the input. Since the purifier is implicitly defined by solving an optimization problem, its purified output is not differentiable with respect to the input data. In other words, $\frac{\partial \text{Purifier}(x)}{\partial x}$ is not well-defined in the same way as neural network purifiers, whose inference stage is fully differentiable.
> > >
> > > This structural property is precisely why applying BPDA is necessary. Under BPDA, when the true backward pass is unavailable, the standard and strongest practice is to replace the backward mapping with an identity mapping. For Pro-Trans, this is the only feasible surrogate, because the purification process does not expose an explicit differentiable computation graph.
> > >
> > > Therefore, treating the purifier as an identity mapping in the backward pass is not a simplification, but the strongest adaptive attack that BPDA can construct in this setting. If an attacker assumes any differentiable surrogate stronger than identity, it no longer correspond to the actual TN-based purification mechanism.

---

### Official Review · Reviewer_2GLB · 2025-10-31

**Soundness:** 3
**Presentation:** 3
**Contribution:** 3
**Rating:** 4
**Confidence:** 4

**Summary:**

This paper addresses the limited generalization of adversarial purification defenses by proposing a novel coarse-to-fine Progressive Tensor Ring method, building upon tensor-based defense strategies. The approach significantly enhances the adversarial robustness of models.

**Strengths:**

This work presents a novel tensor network-based adversarial purification method, described in substantial detail, which significantly improves the computational efficiency of traditional TN approaches in AP tasks.

**Weaknesses:**

1. The defense methods compared in Tables 1–4 vary significantly and lack consistency, which hinders the ability to draw unified conclusions. In particular, it is difficult to assess the specific impact of PTR on SA performance based on the presented comparisons.
2. In Section 4.1, it is mentioned that 512 images were randomly selected for testing. Could author clarify the rationale behind this specific sample size? Additionally, it would be helpful to provide further details regarding the selection process and whether all experiments were conducted exclusively on this subset. The limited sample size raises concerns about potential randomness in the evaluation.
3. Based on the experimental results, PTR appears to suffer from significant overfitting to the distribution of adversarial examples. This is evidenced by a notable improvement in both quantitative and qualitative performance after reconstruction, coupled with a clear decline in the SA metric.
4. I have observed that in Table 5, the performance of the method by Nie et al. shows a noticeable decline on CIFAR-100 compared to its results on CIFAR-10. Could you provide details on how these results were obtained?
5. The ablation study appears to be overly simplistic, as several key components of the method have not undergone thorough ablation analysis.
6. There appears to be an inconsistency in the evaluation of reconstructed image quality. While Figure 5 suggests a noticeable degradation in the visual quality of Pro-Trans reconstructions, Tables 8 and 9 only report quantitative results for the higher-quality PTR outputs. This selective reporting of results makes it difficult to draw meaningful conclusions about the actual performance of the Pro-Trans method.
7. The efficiency analysis appears incomplete, as it only includes comparisons with the TN method. To provide a more comprehensive evaluation, comparisons with other adversarial purification approaches—such as those proposed by Yoon et al. and Nie et al.—should be added.

**Questions:**

See weakness.

---

> ### Author Response · Authors · 2025-11-21
> **Response to Reviewer 2GLB(1/2)**
>
> We appreciate Reviewer 2GLB’s careful assessment and are encouraged by the recognition of the novelty, detailed methodology, and improved efficiency offered by our tensor network-based approach.
>
> Response to Weakness 1:
>
> We understand the reviewer’s concern regarding the inconsistency of the defense methods listed in Tables 1–4. We select baseline methods from RobustBench, and to ensure fairness, we always choose the strongest available methods for the specified dataset, threat, and classifier architecture. As a result, the set of baselines differ across tables. Throughout all experiments, we follow to the standard evaluation protocols widely used in the community, ensuring that Pro-Trans is compared against the most competitive and commonly adopted baselines. The results show that Pro-Trans consistently outperforms existing methods across diverse adversarial settings and achieves state-of-the-art performance. Regarding the effect of PTR on the SA metric, in Table 10, we compare TNP, PTR, Pro-Trans without attention maps, and the Pro-Trans on ImageNet with ResNet-50 under AutoAttack Linf (4/255). The results indicate that PTR achieves the highest SA, which aligns with its design objective of accelerating optimization and improving reconstruction quality but have limited robustness.
>
> Response to Weakness 2:
>
> The choice of a 512-image subset follows the widely adopted protocol in prior optimization-based and diffusion-based purification works[1-3]. The subset was randomly sampled once from the full test set at the beginning of the evaluation, and all experiments in the paper use the same fixed subset to ensure fairness and strict comparability across methods. In addition, to further address the reviewer’s concern, we conducted multiple independent runs and report the mean and standard deviation of the performance, as shown in the table below. The results show that the trends remain consistent with those in the manuscript and do not change our conclusions.
>
> |      |   1   |   2   |   3   |   4   |   5   |   6   | Mean±Std.  |
> | :--: | :---: | :---: | :---: | :---: | :---: | :---: | :--------: |
> |  SA  | 87.69 | 85.35 | 87.69 | 87.69 | 86.13 | 83.59 | 86.35±1.53 |
> |  RA  | 74.02 | 71.48 | 73.82 | 73.63 | 74.8  | 72.46 | 73.36±1.09 |
>
> [1] Nie et al. Diffusion Models for Adversarial Purification. ICLR 2022.
>
> [2] Lee&Kim Robust evaluation of diffusion-based adversarial purification. ICCV 2023.
>
> [3] Li et al. ADBM: Adversarial diffusion bridge model for reliable adversarial purification. ICLR 2025.
>
> Response to Weakness 3:
>
> First, we do not deny that PTR tends to overfit adversarial perturbations to some extent; however, this is in fact a common issue shared by tensor network-based adversarial purification methods in general. In our work, PTR benefits from a progressive optimization strategy that avoids redundant upsampling, which substantially improves the optimization speed and convergence effect of tensor ring and thus enhances its practicality for AP. To specifically address the overfitting issue of TNs, we introduce AGLSR. Regarding the quantitative reconstruction results of PTR, the improved reconstruction quality arises naturally from its better convergence; the qualitative visualizations of PTR’s reconstruction on adversarial examples further show that PTR achieves the smallest reconstruction error, which indeed implies that perturbations are inevitably reconstructed as well. As for the decrease in standard accuracy (SA) of Pro-Trans, this is because AGLSR inevitably discards a certain amount of semantic information, when it is preventing the reconstruction of perturbations. In summary, the improved reconstruction performance of PTR and the SA decrease of Pro-Trans arise from two different module (PTR and AGLSR), and should not be interpreted as evidence that PTR’s reconstruction behavior directly causes the reduction in SA.
>
> Response to Weakness 4:
>
> We thank the reviewer for pointing this out and sincerely apologize for the lack of clarity in our original explanation. Table 5 evaluates cross-dataset generalization, where the AP methods by Nie et al. trained on CIFAR-10 are directly applied to CIFAR-100 without retraining. In this setting, the diffusion model trained exclusively on CIFAR-10 possesses limited generative and purification capacity when applied to CIFAR-100, which lies outside its training data distribution. As a result, when it is used to purify adversarial examples from CIFAR-100, both SA and RA drop significantly. We will update the manuscript to provide a clearer description and deeper discussion of the cross-dataset evaluation protocol and result.

---

> ### Author Response · Authors · 2025-11-21
> **Response to Reviewer 2GLB(2/2)**
>
> Response to Weakness 5:
>
> Pro-Trans consists of three main components: PTR, AGLSR(w/o attention masks), and AGLSR. PTR primarily improves the efficiency and stability of optimization; AGLSR(w/o attention masks) introduces uniform local smoothing via total variation regularization to suppress high-frequency adversarial perturbations, at the cost of some semantic loss; AGLSR further modulates the smoothing strength using classifier-derived attention maps, preserving more details while maintaining the robustness gains from AGLSR(w/o attention masks). And we have conducted an ablation study on ImageNet that isolates the contribution of each part as shown below.
>
> | PTR  | AGLSR(w/o attention masks) | AGLSR |  SA   |  RA   |
> | :--: | :------------------------: | :---: | :---: | :---: |
> |  ✓   |             ✘              |   ✘   | 66.01 | 47.07 |
> |  ✓   |             ✓              |   ✘   | 62.1  | 51.36 |
> |  ✓   |             ✘              |   ✓   | 64.84 | 51.17 |
>
> The results show that PTR alone achieves the highest SA but the lowest RA, reflecting that PTR cannot fully address the problem of reconstruction of adversarial perturbations. Introducing the AGLSR(w/o attention masks) significantly boosts RA by suppressing high-frequency perturbations, at the cost of reduced SA due to the loss of some semantic details. Finally, adding attention masks alleviates this over-smoothing effect, leading to a noticeable recovery of SA while maintaining nearly the same RA.
>
> Response to Weakness 6:
>
> We thank the reviewer for raising this point and would like to clarify that Figure 5 and Tables 8–9 serve different purposes in our evaluation. Figure 5 is designed to illustrate the purification behavior of Pro-Trans, where the goal is not to fully reconstruct an adversarial example. Thus, the observed visual degradation reflects the intended effect of AGLSR in suppressing adversarial perturbations. In contrast, Tables 8 and 9 evaluate a separate question: the reconstruction capability of different tensor networks. These results demonstrate that PTR achieves strong reconstruction performance with fewer parameters. Since the two experiments target different aspects, so indeed they are not contradictory.
>
> Response to Weakness 7:
>
> We appreciate the reviewer’s suggestion and have now included additional efficiency comparisons with other adversarial purification methods. We added the approaches of Nie et al.[1] and Lee & Kim[2], which provide publicly available implementations and support ImageNet dataset. Due to the fact that Yoon et al.’s method does not offer ImageNet support, we were unable to include it in the same setting. As shown in the  table below, we can find that the Pro-Trans have the highest computational efficiency.
>
> |           Methods           | Pro-Trans |  TNP  | Nie et al. | Lee & Kim |
> | :-------------------------: | :-------: | :---: | :--------: | :-------: |
> | Purification time per image |   2.997   | 8.002 |   4.715    |  31.995   |
>
> [1] Nie et al. Diffusion Models for Adversarial Purification. ICLR 2022.
>
> [2] Lee&Kim Robust evaluation of diffusion-based adversarial purification. ICCV 2023.

---

### Official Review · Reviewer_XUrp · 2025-10-31

**Soundness:** 3
**Presentation:** 3
**Contribution:** 3
**Rating:** 6
**Confidence:** 3

**Summary:**

This paper presents Pro-Trans, a tensor network–based adversarial purification framework designed to improve both robustness and computational efficiency. The method introduces a Progressive Tensor Ring (PTR) that performs coarse-to-fine optimization within a fixed tensor topology, avoiding interpolation-based upsampling and its associated instability. In addition, an Attention-Guided Local Smoothing Regularizer (AGLSR) is proposed to adaptively smooth different image regions based on attention maps extracted from the downstream classifier, thereby removing perturbations while preserving semantic information.
Extensive experiments on CIFAR-10, CIFAR-100, and ImageNet show that Pro-Trans consistently improves robust accuracy compared with previous tensor-based defenses such as TNP, while also significantly reducing inference time (about three seconds per image instead of eight).

**Strengths:**

1.	The progressive optimization design within a fixed tensor ring is a thoughtful contribution. It effectively removes the instability and heavy computation of interpolation-based upsampling found in previous coarse-to-fine tensor models. The reasoning behind the design choices is well-motivated and consistent throughout the paper.
2.	Pro-Trans achieves competitive or superior robustness on multiple datasets while maintaining lower computational cost. The convergence analysis and ablation studies provide convincing evidence that PTR improves both stability and speed.
3.	The adaptive local smoothing guided by classifier attention is an elegant way to address over-smoothing and semantic loss. The visualizations clearly show the benefit of using attention masks for targeted denoising.
4.	The experiments are comprehensive, covering different datasets, attacks, and generalization settings (cross-dataset, cross-threat, and cross-attack). The evaluation protocol aligns with the RobustBench standards, and the ablation studies are informative.

**Weaknesses:**

1.	While the method is well-engineered, it still builds directly on the coarse-to-fine tensor purification framework (TNP). The main innovation, namely progressive optimization without upsampling and adaptive smoothing—are important improvements but not entirely new conceptual directions.
2.	The paper lacks a theoretical explanation of why progressive unfreezing of tensor cores leads to faster and more stable convergence. Similarly, the role of attention-guided total variation regularization could be discussed more analytically.
3.	The robustness results are reported on 512-image subsets due to computational cost, which reduces the statistical strength of the conclusions. It would be reassuring to include mean and standard deviation across multiple runs or a larger evaluation sample.
4.	The mathematical sections are dense and assume strong familiarity with tensor network notation. A brief intuitive explanation or a small illustrative figure might make the ideas more accessible to a broader audience.

**Questions:**

•  How are gradients and adaptivity handled in the attention-guided purifier under white-box attacks?
•  How is the progressive optimization schedule chosen, and is it robust to different settings?
•  Does the method generalize to different or unseen classifiers producing the attention maps?

---

> ### Author Response · Authors · 2025-11-21
> **Response to Reviewer XUrp(1/2)**
>
> We thank Reviewer XUrp for the constructive and positive feedback, especially for recognizing the strengths of our progressive TR design, attention-guided smoothing, and comprehensive evaluations.
>
> Response to Weekness 1:
>
> As you pointed out, our method is inspired by the TNP framework, but it differs from it in two key aspects. First, PTR preserves a fixed Tensor Ring topology and achieves coarse-to-fine tensor-network optimization by progressively adjusting both the optimization variables and the optimization objectives, thereby avoiding the interpolation-based upsampling operations constructed via MPO or TT-SVD. This constitutes a fundamental distinction from PuTT-based TNP, where the Tensor Train topology must be prolonged through complex and computationally expensive upsampling in order to realize a coarse-to-fine optimization, leading to slower and less stable convergence. Second, in Pro-Trans, the proposed AGLSR leverages attention maps from the downstream classifier to dynamically assign spatially adaptive local smoothing strengths, in contrast to traditional global, uniform smoothing regularization. Therefore, Pro-Trans can be regarded as a new coarse-to-fine TN-based adversarial purification framework, combining a structurally fixed, progressively optimized TR representation with a semantically aware local smoothing regularizer. We believe this design offers a meaningful extension to prior TN-based adversarial purification methods.
>
> Response to Weekness 2:
>
> Thank you for raising this important point. The goal of progressively unfreezing’ the tensor cores in PTR is to ensure that, at the coarse stage, the optimization target is simple (a low-resolution image) and the optimization variables are limited to only a few cores. This results in a much smaller parameter space and therefore more efficient optimization. When the process moves to finer stages, the solution obtained previously serves as a better initialization for the next stage. In this sense, the coarse stage acts as a warm-up for the finer stage, and PTR becomes a sequence of refinements rather than repeated optimization from scratch. This leads to faster and smoother convergence in practice.
>
> To address the tendency of tensor networks to overfit adversarial perturbations, we introduce AGLSR to guide the purified result toward the natural image manifold. This reduces the sensitivity of the purifier to small perturbations and helps lower the Lipschitz constant of the composite mapping $ \text{Purifier} \circ \text{Classifier} $, which contributes to improved robustness. Moreover, using attention maps from the downstream classifier allows TV to function as a spatially weighted and non-uniform local smoothing regularizer. Regions with strong influence on the classifier’s decision receive stronger penalties on perturbations, while less important regions such as background receive weaker penalties. This achieves a more favorable balance between image quality and robustness.
>
> Response to Weekness 3:
>
> Thank you for your concern regarding the sample size in our experiments. Following recent adversarial purification works, it has become standard practice to evaluate robustness on a randomly sampled subset of 512 test images [1-3]. In line with this established protocol, we also adopt 512-image subsets for our evaluations. In addition, to further address this concern, we conducted multiple independent runs and report the mean and standard deviation of standard accuracy and robust accuracy on CIFAR-10 against AutoAttack Linf (8/255), as summarized in the table below. We observe that the trends are consistent with those in the manuscript and do not alter our original conclusions.
>
> |      |   1   |   2   |   3   |   4   |   5   |   6   | Mean±Std.  |
> | :--: | :---: | :---: | :---: | :---: | :---: | :---: | :--------: |
> |  SA  | 87.69 | 85.35 | 87.69 | 87.69 | 86.13 | 83.59 | 86.35±1.53 |
> |  RA  | 74.02 | 71.48 | 73.82 | 73.63 | 74.8  | 72.46 | 73.36±1.09 |
>
> [1] Nie et al. Diffusion Models for Adversarial Purification. ICLR 2022.
>
> [2] Lee&Kim Robust evaluation of diffusion-based adversarial purification. ICCV 2023.
>
> [3] Li et al. ADBM: Adversarial diffusion bridge model for reliable adversarial purification. ICLR 2025.

---

> ### Author Response · Authors · 2025-11-21
> **Response to Reviewer XUrp(2/2)**
>
> Response to Weekness 4:
>
> We thank the reviewer for the helpful suggestions regarding the tensor-related mathematical notation. Below we provide a more intuitive explanation, which we will further incorporate into the revised manuscript.
>
> | Symbol    | Shape                                           | Description                                                  |
> | :-------- | :---------------------------------------------- | :----------------------------------------------------------- |
> | $X$       | $H \times W \times C$                           | Input RGB image represented as a third-order tensor (height, width, channels). |
> | $X^{(q)}$ | $4 \times 4 \times \ldots \times 4 \times C$    | Quantized high-order tensor representation of the image.     |
> | $G^{(d)}$ | $r_{d-1} \times n_k \times r_d$, with $n_k = 4$ | The $d$-th grid core tensor in the Tensor Ring.              |
> | $G^{(C)}$ | $r_D \times C \times r_0$                       | Channel core tensor.                                         |
> | $r_d$     | Scalar                                          | Tensor Ring rank connecting core $d$ and core $d+1$.         |
> | $Y_d$     | Same as $X$                                     | Reconstructed image tensor at stage $d$ produced by PTR.     |
> | $X_d$     | $2^D \times 2^D \times C$                       | Multi-resolution target tensor at stage $d$, obtained by downsampling and pixel replication. |
> | $I$       | $\text{rank} \times 4 \times \text{rank}$       | Identity core tensor.                                        |
>
> Response to Question 1:
>
> In all our white-box experiments, we apply BPDA to implement adaptive attacks against Pro-Trans and the downstream classifier. Speciffically, during the backward pass of white-box attack, we treat Pro-Trans as an identity mapping. The reason behind is that Pro-Trans does not have fixed parameters, and each input is purified by solving an optimization problem over thousands of iterations, making the purification mapping highly dynamic. Thus, backpropagating through the whole optimization process for every attack step would require storing all intermediate states, which is not practical considering the memory cost and computation time.  Therefore, we apply BPDA to construct adaptive attacks which allow the adversary to adapt to the purifier and avoid gradient obfuscation, thus providing a reliable robustness evaluation rather than overestimating the defense performance.
>
> Response to Question 2:
>
> We primarily follow the optimization schedule used in PuTT and set the total number of iterations for Pro-Trans to 2048. The optimization target and variables are changed at iterations 64, 128, 256, 512, and 1024. We evaluated different optimization schedules and different numbers of stages on ImageNet, and the results are summarized in the table below.
>
> | Schedule |   1   |   2   |
> | :------: | :---: | :---: |
> |    SA    | 64.84 | 67.38 |
> |    RA    | 51.77 | 48.43 |
> | Average  | 58.30 | 57.90 |
>
> |  stage  |   1   |   2   |   3   |   4   |   5   |   6   |
> | :-----: | :---: | :---: | :---: | :---: | :---: | :---: |
> |   SA    | 59.37 | 63.47 | 64.64 | 67.18 | 66.6  | 64.84 |
> |   RA    | 50.19 | 52.92 | 51.36 | 49.60 | 45.89 | 51.77 |
> | Average | 54.78 | 58.19 | 58.00 | 58.39 | 56.24 | 58.30 |
>
> We compared two schedules. Schedule 1 follows the configuration in the main paper. Schedule 2 is a more uniform schedule that also runs for 2048 iterations, with updates at iterations 341, 682, 1023, 1364, and 1705. The results show that Schedule 2 yields higher SA but lower RA, indicating that it restores more perturbations and performs worse than Schedule 1 in terms of robustness.
>
> We also investigated the effect of the number of stages. As the number of stages increases, both SA and RA first improve and then decline. The highest SA is achieved at four stages, and the highest RA is achieved at two stages. Overall, the performance remains stable across most settings. However, when the number of stages is reduced to one, meaning that no coarse-to-fine strategy is applied, both SA and RA degrade noticeably.
>
> Response to Question 3:
>
> Yes, our method does generalize to different and unseen classifiers used to produce the attention maps. In our experiments, we evaluated Pro-Trans with multiple downstream classifiers, including ResNet-50, WideResNet-28-10, and  robust variants of WideResNet-28-10. Across all these architectures, Pro-Trans consistently delivered similar robustness improvements and maintained the same qualitative behavior of the attention-guided smoothing mechanism. This indicates that AGLSR does not rely on the specifics of a particular classifier, but instead relies on the classifier-agnostic property that different networks naturally learn similar patterns of semantically important regions, which allows the attention maps to generalize well across unseen classifiers.

---

### Author Response · Authors · 2025-11-27

Dear Reviewers,

We hope you are doing well. As the discussion phase is approaching its end, we would like to kindly check whether our submitted rebuttal and additional experimental results have adequately addressed the concerns and questions raised in the reviews.

If you feel that any concerns remain unresolved, or that parts of our response would benefit from further clarification or additional experiments, please do let us know. We would be very happy to provide more details or continue the discussion to help the assessment of our work.

Thank you very much for your time, effort, and thoughtful consideration throughout the review process. We sincerely appreciate it.

Warm regards,
The Authors

---

### Author Response · Authors · 2025-12-01
**Summary Comment for Area Chair**

Dear Area Chair,

Thank you for taking the time to evaluate our submission. To help with your assessment, we summarize below how our response addresses reviewers' concerns and why we believe the contributions are meaningful, technically sound, and empirically well supported.

Our work proposes **Pro-Trans**, a novel tensor-based adversarial purification method that integrates **progressive tensor ring (PTR)** with **attention guided local smoothing regularization (AGLSR)**. PTR performs coarse-to-fine optimization within a fixed tensor ring topology, progressively unfreezing cores and optimization targets without any interpolation-based upsampling. This design yields substantially faster and smoother convergence compared to prior TN-based purifiers while preserving the expressive power of the tensor representation. AGLSR uses classifier-derived attention maps to modulate a spatially varying TV regularizer, thereby mitigating the tendency of tensor networks to reconstruct adversarial perturbations while preserving useful details.

We provided more theoretical explanations to clarify the behavior and robustness of Pro-Trans. First, we provide a more detailed intuitive account of why PTR improves convergence: early stages optimize a reduced set of cores against low-resolution targets, effectively lowering the optimization difficulty and turning the full procedure into a sequence of refinements rather than optimization from scratch. We also explain how the local-smoothing prior induced by AGLSR serves to guide reconstructions toward the natural image manifold and reduce the local Lipschitz constant of the purifier–classifier composite mapping, which is beneficial for robustness.

We also clarified that all reported robustness results are obtained under BPDA-based adaptive attacks, where the purifier is treated as an identity mapping in the backward pass. This choice is not a simplification but a necessity. Pro-Trans performs purification by optimizing a low-rank Tensor Ring to reconstruct each input, and does not expose a fixed differentiable forward graph with respect to the input. In this setting, BPDA-identity is the strongest feasible surrogate widely adopted in purification-based defenses, and empirically we observe that Pro-Trans maintains its robustness under this adaptive evaluation.

We also performed extensive parameter sensitivity studies. These include varying the AGLSR weight α, ranks, the number of PTR stages, and different TV forms. Across all settings, Pro-Trans exhibits consistent robustness trends and smooth performance variation, suggesting weak sensitivity to hyperparameters. We also conducted equal-rank comparisons with TNP (e.g., both using rank 10), confirming that Pro-Trans achieves higher robust accuracy. In addition, we provided efficiency comparison against diffusion-based purifiers.

To examine the reliance on attention sources, we evaluate Pro-Trans with attention maps generated by different, mismatched classifiers while keeping the downstream classifier fixed. The resulting clean and robust accuracies degrade only slightly, indicating that AGLSR leverages generic patterns of semantic feature rather than depending on a specific architecture. Finally, following standard practice in diffusion-based AP works, we justify the use of a 512-image test subset with multiple independent runs, showing low variance in both standard and robust accuracy and stable ranking among methods.

Taken together, these results support our main claims: Pro-Trans provides a practically efficient and empirically robust tensor network–based adversarial purification framework, with clear analytical intuition and thorough experimental validation. We sincerely appreciate your time and careful assessment under the current challenging circumstances.



Best regards,

Authors.

---

### Meta-Review · Area_Chair_cevR · 2026-01-13

**Summary:**

The paper proposes Pro-Trans, a tensor-based adversarial purification framework that combines a Progressive Tensor Ring (PTR) optimization with an Attention-Guided Local Smoothing Regularizer (AGLSR). By performing coarse-to-fine optimization within a fixed Tensor Ring, the method significantly improves computational costs and convergence stability over prior TN-based purification methods such as TNP. The attention-guided regularization further mitigates over-smoothing by preserving semantically important regions. In experimental results, Pro-Trans demonstrates notable gains in robust accuracy with lower computational costs compared to TNP.

Reviewers agreed that the efficiency and empirical robustness improvements are strong. The rebuttal substantially strengthened the paper by adding hyper-parameter sensitivity analyses (rank, stages, \alpha), mismatched-attention experiments, multiple-run statistics on the 512-image subset, and efficiency comparisons against diffusion-based purifiers.

However, several core concerns remain only partially resolved. An issue raised across reviewers   (ZYNP, Lu4h, XUrp) is that the main contributions (PTR and AGLSR) are viewed more as well-engineered extensions of existing coarse-to-fine tensor purification frameworks rather than as a fundamentally new concept. While the rebuttal provides intuitive explanations (e.g., the coarse stage acts as a warm-up and attention-weighted total variation acts as a robustness prior), the paper still lacks a deeper theoretical or analytical understanding of why these design choices lead to improved robustness beyond empirical observation.

More importantly, the evaluation under strong adaptive attacks remains controversial. The authors clarified why they adopt BPDA with an identity backward pass for their optimization-defined TN purifier. However, Reviewer ZYNP remains unconvinced that this evaluation captures the strongest possible adaptive attacks, leaving some residual uncertainty about robustness claims.

In addition, although many experimental concerns were addressed in the rebuttal, reviewers continued to raise concerns about the clean–robust trade-off. In particular, multiple reviewers noted that PTR tends to reconstruct adversarial perturbations (2GLB), while AGLSR improves robustness at the cost of reduced the SA metric (2GLB, Lu4h). While the authors explained this behavior as an inherent trade-off of adversarial purification, some uncertainty remains regarding how this trade-off should be interpreted in practice.

Overall, the paper presents a technically solid and practically promising improvement to tensor-network–based adversarial purification, with strong results and much improved efficiency. Nevertheless, due to the remaining concerns regarding conceptual novelty, lack of theoretical insight, and the strength of the adaptive attack evaluation, the AC recommends rejection at this time.

**Reviewer Concerns:**

Addressed: XUrp
Not fully addressed: 2GLB, Lu4h

**Reviewer Scores:**

XUrp: 6->6
2GLB, Lu4h: 4->4

---

### Decision · Program_Chairs · 2026-01-26

Reject